# UNABLE TO FORGET: PROACTIVE INTERFERENCE REVEALS WORKING MEMORY LIMITS IN LLMS BEYOND CONTEXT LENGTH

## ABSTRACT

Information retrieval in Large Language Models (LLMs) is increasingly recognized as intertwined with generation capabilities rather than mere lookup. While longer contexts are often assumed to improve retrieval, the mechanics of intra-context interference, as instantiated in MRCR test, remain understudied. To address this, we adapt the proactive interference (PI) paradigm from cognitive science, where earlier information disrupts recall of newer updates. In humans, susceptibility to such interference is inversely linked to working memory capacity. We introduce PI-LLM, an evaluation to measure LLM working memory by sequentially streams co-referenced key–value updates, where the same key is sequentially rebound to multiple values, and queries only the final values. Although these final values are clearly positioned just before the query, LLM retrieval accuracy declines log-linearly toward zero as co-referenced interference accumulates; errors arise from retrieving previously overwritten values. Attempts to mitigate interference via prompt engineering (e.g., instructing models to ignore earlier input) yield limited success. These findings reveal a fundamental constraint on LLMs' ability to disentangle interference and flexibly manipulate binding information, suggesting a working memory bottleneck beyond mere context access.

PI-LLM bridges (i)LLM performance in MRCR tests and (ii) studies of entity binding in LLM mechanistic interpretations. And provides a cognitive-science inspired measurement of LLM working-memory-like capacity.

## 1 INTRODUCTION

Current research indicates that Large Language Models (LLMs) generally struggle with retrieval tasks when closely related pieces of information are present (Vodrahalli et al., 2024). Furthermore, reasoning models do not effectively improve performance in these scenarios (OpenAI, 2025b). However, most studies—having already labeled retrieval as a 'long-context' challenge—prioritize input length as the primary determinant of retrieval difficulty, relegating other factors to a secondary role.

Current studies often conflate search difficulty—the challenge of locating the relevant "needle" in a vast contextual haystack—with interference—the challenge of correctly identifying that needle when it is surrounded by similar-looking but incorrect items. Recent long-context benchmarks—most of which evolve from the original Needle-in-a-Haystack paradigm, such as DeepMind's Michelangelo (Vodrahalli et al., 2024) and OpenAI's MRCR (OpenAI, 2025c) primarily raise task difficulty by lengthening the prompt. Although these studies acknowledge interfering information's impact on the retrieval tasks, they do so only in a preliminary way, without explicitly isolating or quantifying interference's independent effect on LLMs' context usage. Consequently, current research implicitly attributes the difficulty of distinguishing similar information mainly to greater input length, thereby overlooking interference as a separate, quantifiable factor.

Our work demonstrates that the amount of interfering information—Coreferenced information—independently and significantly impacts retrieval accuracy in LLMs (Figure 2). By systematically varying interference load, we obtain the first quantitative curve that isolates interference as an independent factor. To demonstrate that interference effects are independent of input length, we include a control condition in which input length is held constant. Anti-interference capacity varies

sharply across models, making it a useful discriminative trait. Crucially, even modest distractor loads expose a fundamental weakness: current LLMs cannot reliably suppress competing cues.

Interfering information consists of Co-referenced information and is common in many data processing tasks. One of the simplest forms involves key–value pairs, where the key remains the same but the associated value is repeatedly updated within a sequence. For example, consider a sequence of blood-pressure (BP) readings, where the task is to keep track of the most recent BP value. BP: 120 – triage; BP: 128 – 10 min later; BP: 125 – discharge. In this task, the desired output is 'BP: 125,' the last-presented key–value pair. However, retrieval may be impaired by prior semantically similar BP values, which act as distractors. The search difficulty in such key-value tracking tasks is minimized, as the target answer is always the last value of a certain key.

Notably, humans demonstrate high accuracy on these tasks. In contrast, our experiments show that **retrieval accuracy in state-of-the-art LLMs declines in a log-linear fashion as the amount of interference information preceding the target key–value pair increases**, as shown in (Figure 2), a pattern we observed consistently across all models tested.

While standard synthetic key–value retrieval tasks are widely used in LLM evaluations (e.g., Lost in the Middle), our approach uniquely leverages insights from the proactive interference (PI) paradigm in cognitive psychology. In classic PI experiments, participants recall the most recent association for a repeated cue while earlier associations cause interference. Drawing from the PI paradigm, we fix the retrieval target as the last-presented value of a particular key, thereby minimizing search difficulty and isolating interference as an independent factor. We ensure this by explicitly prompting LLMs to retrieve the most recent key–value pair for a given key. We systematically manipulate the amount of Co-referenced interfering information preceding the target and measure the effect on retrieval accuracy. This approach allows us to directly quantify the impact of interference strength, independent of search difficulty.

Surprisingly, we found that a higher interference load leads to a **log-scale reduction** in retrieval accuracy **even when input length remains constant**, revealing that **input length and interference are independent** factors affecting retrieval. Moreover, we observed that log-linear declines occur across multiple interference dimensions (e.g., increasing the token length of the retrieval target), consistent with the idea that LLM retrieval is limited by a **unified capacity**—a resource that can be exhausted along any one dimension, or conserved by reducing load along another. This shared bottleneck closely mirrors the working memory limit observed in humans.

Cognitive science research on proactive interference (PI) shows that, although humans are also affected by prior interference information, their recall performance typically plateaus: after a certain threshold, further interference produces minimal additional impairment. This robustness is attributed to humans' ability to actively unbind outdated associations from working memory before encoding new information (Oberauer & Vockenberg, 2009).

Building on these observations, we further investigated whether LLMs could adopt human-like strategies for managing interference through explicit modulation of memory content. Humans benefit from direct instructions to deprioritize prior interfering information (Festini & Reuter-Lorenz, 2014). To test whether similar explicit strategies could aid LLM performance, we provided natural language annotations marking the majority of prior information as outdated and irrelevant. Despite clear instructional cues and explicit annotations, we observed only minimal improvements in LLM retrieval accuracy.

We evaluated **reasoning models** with an unlimited reasoning budget and observed the **same log-linear decline** in retrieval accuracy. We further injected explicit chain-of-thought (CoT) prompts into non-reasoning models, instructing them to first analyze the task goal and then retrieve the last value. However, this intervention yielded no improvement over the baseline; **despite producing extensive reasoning tokens**, retrieval performance continued to decline log-linearly.

Our findings can be distilled into the following points.

- **Interference overrides recency and instruction.** Interfering information consistently and substantially degrades LLMs' ability to retrieve target content. Errors are dominated by the retrieval of prior co-referenced values, even when the correct answers are unambiguously located near the end of the input.

- **Universal log-linear decay.** Across all SOTA models in our study, retrieval accuracy declines in a clear log-linear fashion toward zero with 3 dimensions: increasing update count (Figure 2), number of tracked keys (Figure 4), and value length (Figure 14). These results suggest a consistent negative log-linear relationship between retrieval performance and information load.

- **Marginal effectiveness of reasoning models and natural language prompt interventions.** LLMs are capable of articulating the correct retrieval procedure, yet they consistently fail to implement it in execution when under interference. These findings reveal a **dissociation between analytical reasoning and execution**.

## 2 INTERFERENCE DOMINATES RETRIEVAL DESPITE RECENCY AND INSTRUCTIONS:

Our objective is to understand how Large Language Models (LLMs) manage interference when retrieving information. To reduce searching difficulty and measure the impact of interference, we designed a synthetic key-value retrieval experiment.

In this test, the input is a sequence of key–value pairs, where a fixed set of keys—each representing a variable of interest—appears repeatedly throughout the sequence, each time paired with a different value. Updates for different keys are randomly interleaved. This design mimics, in a simplified manner, real-world logging systems that track multiple physiological variables over time—for example, blood pressure, heart rate, and oxygen level readings recorded in a patient's health log.

### 2.1 EXPERIMENTAL DESIGN

Figure 1: Basic input example for LLM Proactive Interference (LLM-PI) test. In this example, three keys ("visual art", "tools", and "landform")—color-coded for clarity—each undergo four updates. In the actual experiment, up to 46 keys were used, each updated up to 400 times with distinct values. For visual clarity, numerical prefixes (e.g., "1*") were added to show the update order, but these were not present in the input. The model is instructed to retrieve the final value for each tracked key, indicated in bold for illustration. The keys to check are cued both before and after the update stream.

In this task (Figure 1), each input sequence consisted of three parts: 1. Instruction—a brief directive indicating the task and specifying which keys to track for value updates. 2. Update stream—a sequence of key–value pairs, where a fixed set of keys each receive an equal number of updates. The updates for different keys are randomly interleaved and are organized such that the same key does not appear in two consecutive key-value pairs. 3. Query—a prompt instructing the model to retrieve the final value associated with each tracked key.

The retrieval objective was to return the most recent value associated with each specified key. For each key with update count X, the preceding X–1 key–value pairs served as irrelevant, interfering updates—sharing the same key but differing in value. This design allowed us to isolate the effect of interference.

Because the retrieval target is always the value from the last occurrence of each key, search difficulty is ideally low: the model simply needs to locate the most recent update for each key. While the random interleaving of updates does not guarantee that the last occurrence of each key is near the end of the sequence, the retrieval target's relative position is always clearly defined as the most recent appearance. As a result, the search space is small and well-informed. The main challenge, therefore, is not finding the target, but correctly identifying it in the presence of multiple earlier, competing updates. This mirrors realistic data environments where many variables are updated concurrently, and interference—rather than search—becomes the limiting factor.

In this particular experiment, we used 46 unique keys, each receiving multiple value updates throughout the sequence. For each key, the last value it receives is the retrieval target, while all earlier key–value pairs for that key—totaling 46 × (update count – 1) interfering distractors—serve to induce interference. Figure 1 provides an example input and its corresponding output for three keys undergoing multiple updates. We measure accuracy by counting the number of correctly retrieved final values across all keys.

This synthetic key–value retrieval task is related to "Lost-in-the-Middle" (Liu et al., 2024), which examines how the position of the retrieval target within the context affects accuracy. In contrast, our approach offers finer experimental control over interference: by always probing the most recently updated value for each key, we hold the target's relative position constant. In later experiments, we also fix the total input length, allowing us to systematically isolate and measure the effects of interference in the retrieval task.

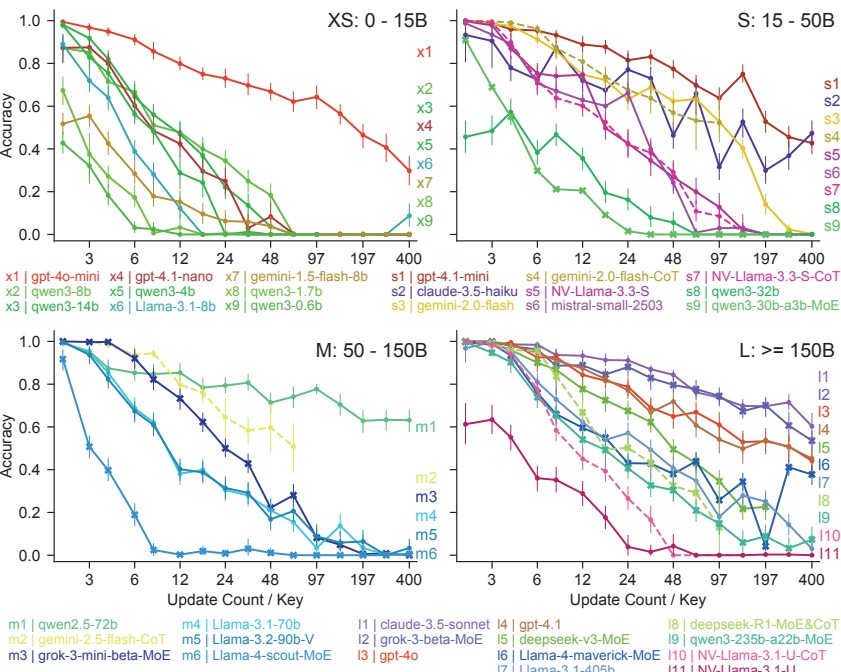

Figure 2: Universal *log-linear* decline in retrieval performance due to interference. Increasing the amount of interfering information preceding a retrieval target within a language model's input context results in a log-linear decrease in retrieval accuracy across diverse models. The target is positioned after the interfering information and explicitly referenced in the prompt to reduce search difficulty and isolate interference effects. (x-axis: the number of Co-referenced information, log-scaled; asterisk: MoE models).For visualization, models were grouped by estimated parameter size into four tiers—XS, S, M, and L—shown from top to bottom. Larger models (L group) tended to degrade more slowly, while smaller models (XS group) declined the fastest.

## 2.2 RESULTS AND DISCUSSION

**Interference information severely impairs the ability of LLMs to effectively utilize context information.** Across models of varying parameter sizes, we observe a robust log-linear decline in retrieval accuracy as additional interfering key–value pairs are inserted before the target value for each key (Figures 2). This log-linear trend reflects rapid initial accuracy loss, with subsequent interference causing smaller additional declines. Notably, the log-linear effect persists across models of different developmental stages and model sizes; larger models exhibit a more gradual decline than smaller ones.

**Robustness: Prompt and Sequential Mode.** We validated robustness with additional prompt variants and by switching between random and sequential modes; across all variants, the declining trend persisted (Appendix C; Figure 10). Each evaluation used freshly sampled input sequences, and we report 95% confidence intervals via nonparametric bootstrapping for all tests in this paper.

## 2.3 INCORRECT EXTRACTIONS ARE PRIMARILY ATTRIBUTED TO PROACTIVE INTERFERENCE

LLM extraction errors increase consistently when interference information is present, and analysis reveals that most errors are from earlier, outdated key-value pairs—mirroring proactive interference (PI) in cognitive science, where previously learned information hinders the retrieval of more recent information.

We observed a systematic shift in the distribution of model outputs as interference increased as shown in Figure 3 (see also Figure 22 for more detailed examples). Across increasing interference, LLM retrieval errors displayed a systematic three-stage evolution. **Stage 1—at low interference**, errors were rare and highly localized to the most recent updates for each key. ("recency errors"). **Stage 2—as interference accumulated**, retrieval accuracy rapidly dropped towards zero; errors shifted further back in the update history, with the model increasingly favoring earlier updates. The growing temporal span of the error distribution marked a clear drift from recency towards primacy. **Stage 3—as interference further intensified**, retrieval accuracy remained near zero. The error distribution evolved from a dispersed pattern to a tight anchoring on the earliest updates, signifying a complete dominance of primacy effects. Simultaneously, hallucinations—the output of values never present in the update history—also increased.

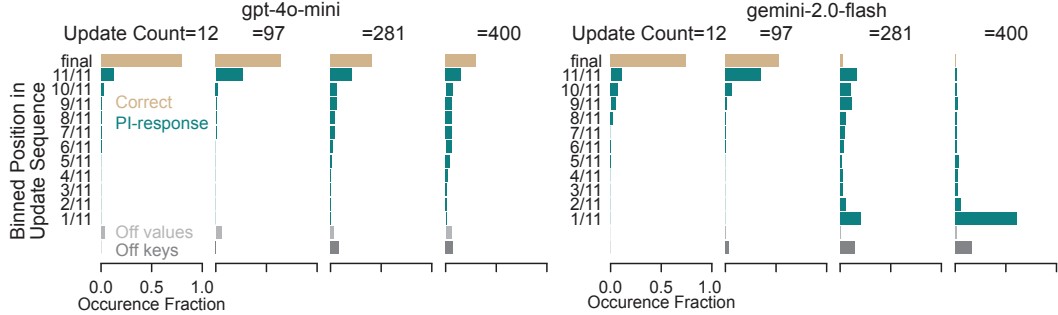

Figure 3: Distribution of model responses across update positions, showing increasing signs of PI as update count increases (left to right). The y-axis lists 11 equal-width bins (Bin 1–Bin 11, green) covering the entire update sequence. The earthy yellow bar indicates the single final update—the correct retrieval target. Light gray bars ("off values") denote cases where the model returns a value not present in the update history (i.e., hallucinations). Dark gray bars ("off keys") indicate failures to return any value for the queried key. As update count increases, errors shift from clustering near the final update to earlier bins, with rising rates of off-values and off-keys. For response distributions from additional models, see Figure 22 in Appendix.

This distribution change aligns with our "limited resource" hypothesis. Specifically, we observed a migration in error distribution: as interference increased, models shifted from making localized recency errors to being strongly anchored in early updates. Even when retrieval accuracy was near zero, this capacity continued to be consumed, causing the model's responses to drift progressively further from the correct values.

In cognitive science, PI resilience is highly correlated with human working memory capacity. Our results suggest that an LLM's anti-interference capability could serve as a metric for its working-memory-like capacity—not merely the ability to store information, but also the capacity to actively maneuver and manage it. However, whether this process is continuous or involves distinct phase transitions requires further investigation.

We further defined an Interference Endurance Score (IES) to quantify each model's resistance to interference (Figures 19 and 20). Regression analyses show that model parameter size is predictive of IES, whereas context window length is not. This suggests that, much like working memory in humans, an LLM's resilience to interference depends more on its underlying computational resources (parameter size) than on the sheer amount of information it can process (context window). See Figure 8, Table 1 in Appendix B.0.2 for more details. **Mixture-of-Experts (MoE) models tend to underperform** dense models with similar total parameters, likely because only a subset of parameters is active per forward pass (see Figure 9 in Appendix B.0.2)

## 3 INTERFERENCE IS INDEPENDENT OF INPUT LENGTH

Retrieval accuracy in language models declines log-linearly as the update count per key increases, suggesting a limited working-memory-like capacity. However, in the previous experiment, input length was not controlled; thus, the observed decline might simply reflect increasing context length rather than genuine interference. To directly test the role of interference, we designed two additional experimental settings.

### 3.1 EXPERIMENT SETUP

1. **Settings A – Number of Updated Keys** ($N_U$): We fixed the update count for each key and increased interference by varying the number of distinct keys updated in the sequence ($N_U$, from 2 to 46). This contrasts with our earlier experiment, which held the number of keys constant while varying the update count per key.

2. **Settings B – Number of Tracked Keys** ($N_T$) **at Fixed Input Length**: In this condition, both the update count per key and the number of updated keys ($N_U$) were fixed, so each input sequence contained the same number of key–value pairs. However, we varied the number of keys the model was instructed to track and retrieve at the end—these are the tracked keys ($N_T$), chosen from among the $N_U$ updated keys, with $2 \leq N_T \leq N_U$. Figure 11 in the appendix provides an example: among 3 distinct keys updated in the sequence ($N_U = 3$), only 2 are tracked ($N_T = 2$) and queried at the end.

By manipulating interference both with and without changes in input length, we can dissociate the effects of interference from those of context length; observing similar declines in retrieval accuracy across both settings would provide strong evidence that interference, rather than context length alone, constrains model performance. Input example illustrating how the model is prompted to track and return values for a subset of updated keys in the appendix. Figure 11.

### 3.2 RESULTS FOR BOTH SETTINGS

Both Settings A and B **exhibit nearly identical** log-linear declines in retrieval accuracy, as shown in Figure 4.Notably, **Setting B kept input length fixed while Setting A allowed it to grow.**

- **Left panel:** Number of *updated keys* in **Setting A**
- **Right panel:** Number of *tracked keys* in **Setting B**

This similarity indicates that **the observed performance drop cannot be attributed solely to longer input sequences; rather, it is driven by increased interference from tracking more keys.**

Furthermore, models that excel in the variable input length setting (Setting A) also performed well in the fixed-length setting (Setting B), underscoring the robustness of this pattern across experimental setups. **The universal log-linear decline observed, even under fixed input length, suggests that anti-interference capacity operates as a distinct resource, independent of the total input length.** See Appendix D for a more detailed analysis.

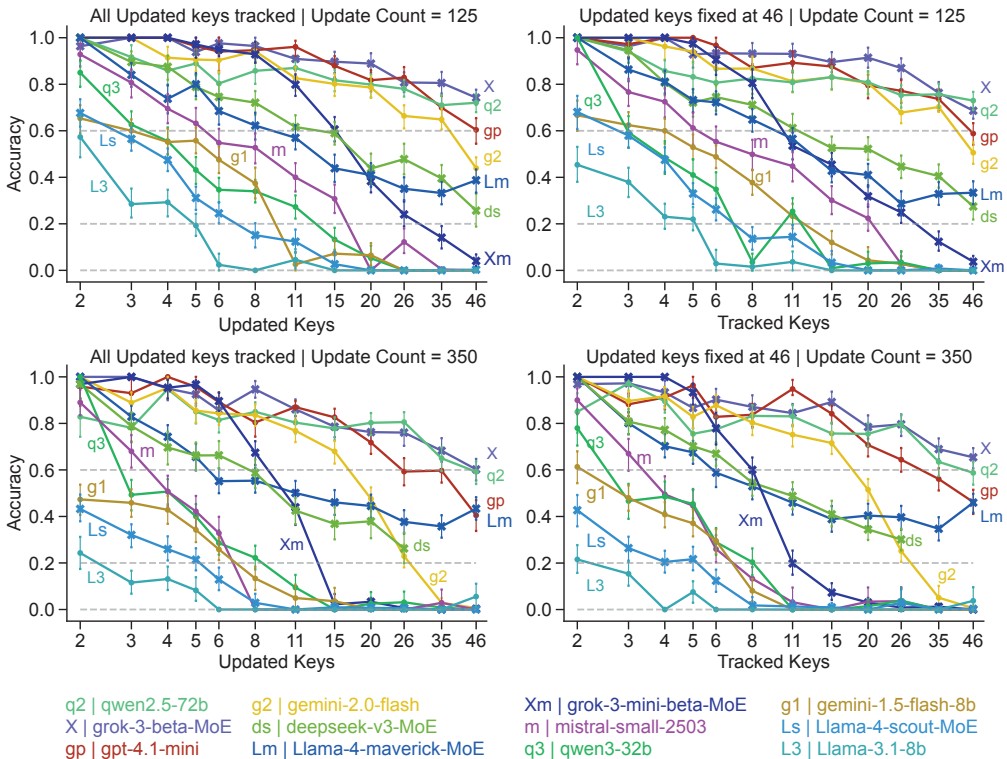

Figure 4: Varying the number of updated keys (**left panels**) versus the number of tracked keys (**right panels, with updated keys fixed at maximum**) yields only minor differences in retrieval accuracy. In all conditions, accuracy declines approximately log-linearly with the number of keys. Each key is updated a fixed number of times—125 in the upper panels and 350 in the lower panels. Some models exhibit a two-phase decline; MoE models are indicated by "X" markers. Error bars represent 95% confidence intervals computed via bootstrapping. Model acronyms are used to label the corresponding curves.

## 4 RETRIEVAL CAPACITY IS LIMITED BY A SINGLE INTERFERENCE BOTTLENECK ACROSS DIMENSIONS

Additionally, we observed **similar log-linear declines** in retrieval accuracy when **manipulating other dimensions** of information load, such as increasing the **token length** of the value in the key-value pair. The analysis of these different Settings implies that the **LLM's capacity to resist interference is Limited by a Single Interference Bottleneck Across Multiple Dimensions,** paralleling findings on human working memory capacity. (Baddeley et al., 1975)

A comprehensive analysis and corresponding performance graphs are provided in Appendix E, which further illustrate the log-linear decline pattern emerges again as token length increases (see Figure 14).

## 5 MITIGATING INTERFERENCE

Evidence from cognitive science indicates that humans are capable of actively unbinding prior associations before encoding new information (Oberauer & Vockenberg, 2009). We hypothesize that LLMs lack such unbinding mechanisms, which explains their continuous monotonic decline in retrieval accuracy as interference increases—ultimately dropping to 0%, indicating complete retrieval failure under high interference conditions.

We therefore evaluated whether LLMs could adopt human-like mechanisms to manage their working memory content explicitly or implicitly in response to interference. Specifically, we tested in-

terventions with prompts instructing the models either to explicitly "forget" prior associations or to implicitly disregard irrelevant information by marking certain prior updates as outdated. However, these strategies yielded minimal improvement, revealing that current LLMs lack the ability to effectively translate either explicit or implicit forgetting instructions into genuine enhancements in retrieval accuracy.

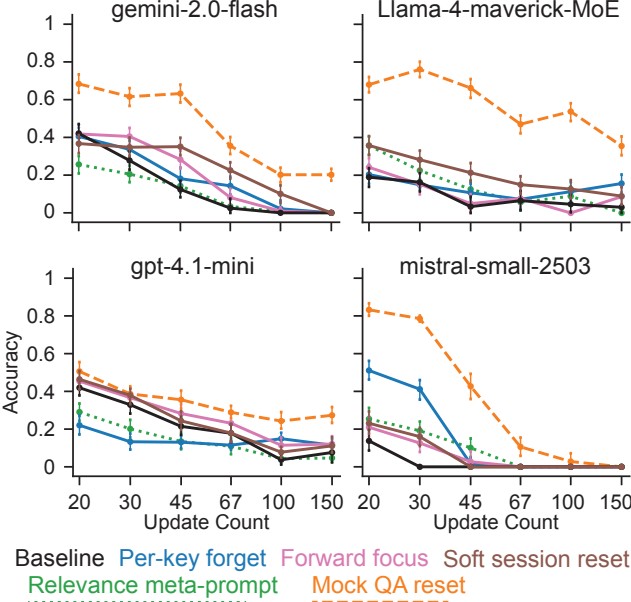

Figure 5: Explicit forgetting and focusing prompts inserted during the update stream (as shown in Figure 15 yielded only marginal improvements in retrieval accuracy. The black line indicates the baseline condition with no intervention prompt. Solid lines represent several simple natural language prompts designed to instruct the model to forget previous updates, focus on upcoming ones, or reset context. For most models, these interventions had limited effect, especially at higher update counts, where the baseline performance is low. The per-key forget even had a negative effect on gpt-4.1-mini. The relevance meta-prompt (green dotted), which asked the model to self-assess what to focus on, was ineffective for all models and even harmful for gpt-4.1-mini. Only the mock QA reset intervention (orange dashed line), which simulates a user-model interaction, led to a substantial improvement in retrieval accuracy. However, this strategy was not immune to the overall trend: accuracy continued to decline with increasing update count (log-spaced).

**Natural Language Interventions fail to relieve interference**

(i) Attempts to mitigate interference through natural language prompts (Figure 15)—whether by explicitly marking information as outdated, instructing the model to "forget" earlier updates, or emphasizing newer information—consistently prove ineffective. Across diverse prompt types and experiment settings, retrieval accuracy shows little to no improvement and forget prompts can even reshape errors toward the injection point (i.e., models preferentially pick values just before the "forget" cue). See Figure 5 for accuracy under each prompt and Figure 17 for the error localization pattern; full prompt designs appear in Appendix F. These results indicate the robustness of the interference effect and its resistance to standard language-based prompt interventions. (ii) A "hack" prompt. Inspired by LLM "hacking" studies showing that models can be coaxed to bypass earlier instructions (Kuo et al., 2025), we devised a non-natural-language prompt that coaxed the model to treat preceding input as belonging to an already processed prior task, thereby partially mitigating interference and improving retrieval accuracy. This ad-hoc "reset" lifts retrieval accuracy across models (orange dashed line in Fig. 5); however, the overall log-linear decay persist. Full design and extensive test details are in Appendix F.3.

## 6 WHY REASONING MODELS FAIL TO IMPROVE RETRIEVAL: TOP DOWN VS BOTTOM UP

LLMs were susceptible to interference across all tested prompts. To eliminate potential ambiguities in task instruction, we designed a CoT style—"activate-locate" prompt (full prompt in Relevance meta-prompt in Figure 15), which first required the LLM to **analyze and state the location of the target** key-value pair within the input. While **models could correctly identify that the answer was at the very end**, this knowledge did not translate into improved retrieval performance; accuracy remained comparable to baseline conditions and **still exhibited a consistent decline** (Figure 5).

We evaluated various models alongside their reasoning counterparts:(Deepseek V3,R1), (Gemini Flash 2.0, Gemini Flash 2.5 with Reasoning on) and models offering both Reasoning and non-Reasoning mode (such as Nvidia-Llama). Consistently, we observed that the **latency and cost** of reasoning models are significantly higher than non-reasoning models on this retrieval task, yet **reasoning models do not improve performance in this test** (Figure 6).

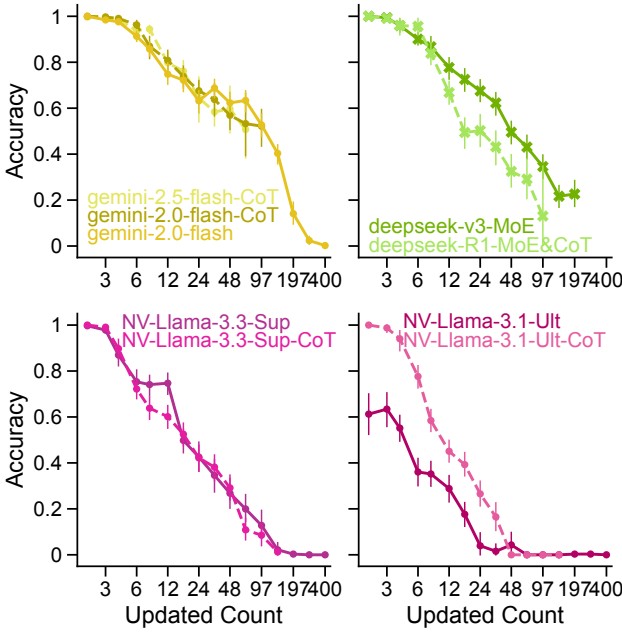

Figure 6: Reasoning Models do not improve retrieval performance. Accuracy as a function of update count is shown for four pairs of Reasoning (CoT) models and their corresponding base (non-CoT) versions. In three of the four comparisons, the CoT variant performs worse than or equal to its base model. CoT variants are only tested up to about 100 updates or less due to output overflow ("thinking" exhaustion). Solid lines denote base models; dashed lines denote (Chain of Thought)Reasoning models. The x-axis is log-scaled.

This reveals a **discrepancy between an LLM's top-down analytical reasoning and its bottom-up information processing** and retrieval execution: knowing "where" the answer is does not translate into its ability to retrieve it under interference. This gap highlights the absence of top-down executive control in guiding retrieval behavior.

## 7 MECHANISTIC INTERPRETATION

Prior Work on RAG has established that intra-context conflicts degrade LLM performance (Lee et al., 2025; Wang et al., 2025; Zou et al., 2025; Chen et al., 2022). (Zhou et al., 2025)

Our focus is more *mechanistic*. PI-LLM isolates a specific configuration where the same key or entity is repeatedly rebound to different values (A→$B_1$, A→$B_2$, ..., A→$B_n$), and the model must retrieve a particular binding/reference. Our experiment connects directly to entity-binding inter-

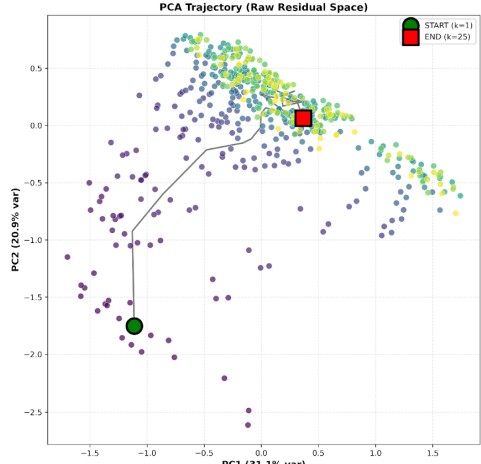

Figure 7: PCA representations in Llama-3.2-1B (Layer 10). We construct a PI-LLM test where only a single key (key1) is sequentially updated to 25 distinct values ($N = 25$). For each query of the form "What is the $n$-th value of key1?" ($n = 1, \ldots, 25$), we extract the residual-stream activation at the last input token (just before generation) in Layer 10 and run PCA on the resulting 25 vectors. The plot shows the first two principal components of these activations: each point is a single query representation, colored by which value index it targets (1st, 2nd, ..., 25th value). This figure provides an introductory, single-layer view of how different value indices for the same key are organized within a low-dimensional subspace of the residual stream.

pretability work (Dai et al., 2024; Feng & Steinhardt, 2024). Prior work shows that when a model encodes bindings such as 'A associated with B', its internal representations occupy a structured subspace from which both the identity of A and its bound value B are decodable.

We run a mechanistic probe of PI-LLM and observe a binding geometry that parallels prior entity-binding results: a single key is updated 25 times within one context, and we then issue queries of the form "What is the $n$-th value of key1?" for $n = 1, \ldots, 25$. For each query, we extract the residual stream at the last input token (just before generation) at every layer of Llama-3.2-1B and perform PCA per layer over the 25 query representations. **(Figure 7) show that different referenced values for the same key occupy meaningfully separated regions of representation space**, with **progressively more compression and overlap** as the number of bindings grows.(We also conduct **full layer scan. See (Figure 24) in appendix)**

MRCR has become a standard long-context evaluation for SOTA. models such as Gemini 3 Pro and GPT-5 (Google DeepMind, 2025; OpenAI, 2025a). MRCR test show that language models can fail to retrieve the correct value once an entity has been referenced(repeatedly associated to new values) multiple times, even though all necessary information is present in context.

PI-LLM bridges real-world MRCR benchmarks and the entity-binding of LLM interpretability.

PI-LLM serves as a minimal abstraction of MRCR that removes the haystack/search components and isolates the multi-coreference binding–unbinding core. PI-LLM provides a quantitative measure of the underlying difficulty.

## 8 CONCLUSION

PI-LLM bridges between (i) MRCR performance, (ii) LLM mechanistic studies of entity binding and calls for approaches that strengthen models' ability to handle binding-related retrieval tasks.

We propose LLM's working memory–like capacity can be measured by its ability to resist interference of multiple bindings. Our experiments show that LLMs share several key constraints with human working memory.

REPRODUCIBILITY

The source code for all experiments can be found in the supplementary material and will be publicly released.

We also provide a static, versioned dataset snapshot (for quick verification) generated from our code; it is included in the supplement. For speed, this snapshot reports the mean over 10 fixed stationary sessions and omits 95% confidence intervals—this differs from the full evaluation pipeline, which computes 95% CIs over repeated runs.

ETHICS STATEMENT

This work evaluates large language models on synthetic key–value tracking tasks. No human subjects, personal data, or sensitive real-world data were used.

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

# A APPENDIX

## A.1 NOTES

All experiments were concluded by May 5th, 2025. For a detailed list of model versions, please refer to Appendix I.1.

## A.2 THE USE OF LARGE LANGUAGE MODELS

We used large language models solely for proofreading (typos, grammar).

### A.3 FULL DETAILED EXPERIMENTAL OVERVIEW

**Experimental Overview**

Our experiments systematically demonstrate that interference is among the primary factors limiting retrieval accuracy in LLMs. **Experiment 1** reveals a robust, log-linear decline in accuracy as interference increases. **Experiment 2**, which holds input length constant, confirms that this effect is driven by interference itself rather than input length. **Experiment 3** further shows that retrieval performance is universally constrained by an interference capacity limit, which can be lated in multiple ways. **Finally**, we investigate mitigation strategies, offering new insights into LLMs' ability to manage in-context information under interference.

## B INTERFERENCE DOMINATES RETRIEVAL DESPITE RECENCY AND INSTRUCTIONS:

Our objective is to understand how Large Language Models (LLMs) manage interference when retrieving information. To reduce searching difficulty and isolate the impact of interference, we designed a synthetic key-value retrieval experiment.

**Data and Performance Evaluation**

To maintain comparability with human performance, we constructed a word dictionary with up to 46 categories, each comprising 400 words. The token lengths of words within each category were selected to fall within a similar range. Keys were drawn from these category names, and values were randomly selected from the corresponding categories in the dictionary. This dictionary design aligns with cognitive psychology proactive interference tests related to human working memory. Words were randomly selected from the dictionary in each test run to eliminate the potential effects of specific semantic combinations. Confidence intervals (CI95) were computed using bootstrap methods after multiple test repetitions.

This synthetic key–value retrieval task is closely related to "Lost-in-the-Middle" (Liu et al., 2024), which examines how the position of the retrieval target within the context affects accuracy. In contrast, our approach offers finer experimental control over interference: by always probing the most recently updated value for each key, we hold the target's relative position constant. In later experiments, we also fix the total input length, allowing us to systematically isolate and measure the effects of interference in the retrieval task.

**Models**

We evaluated a broad spectrum of state-of-the-art open-source and proprietary LLMs, ranging from 0.6B (Qwen3-0.6B) to 637B parameters (Deepseek-V3), and including major proprietary models such as GPT, Claude, Gemini, and Grok. Our benchmark covers both dense and Mixture-of-Experts (MoE) architectures, spanning diverse training data volumes and hardware resources.

### B.0.1 INCORRECT EXTRACTIONS ARE PRIMARILY ATTRIBUTED TO PROACTIVE INTERFERENCE

For response distributions from additional models, see Figure 22

Given the consistent decline in LLM extraction accuracy when interference information is introduced, we investigated the underlying causes of these errors. Our analysis of the input sequences that appeared in the LLM's responses reveals that **errors are predominantly influenced by information encountered before the final, correct update** to a given key. This phenomenon is analogous to proactive interference (PI) in cognitive science, where previously learned information hinders the retrieval of more recent information.

We observe a three-stage progression in error distribution patterns as interference increases:

**Stage 1 – Low Interference, Tightly Focused Errors**: When interference is low, retrieval accuracy is high and the model's error distribution is sharply peaked around the correct value. Errors, when they occur, are not random but show a consistent pattern: they tend to be earlier key–value pairs for the same key, typically located in positions (bins) immediately preceding the final, correct value.This indicates that the model's confusion is narrowly constrained and spatially localized.

**Stage 2 – Moderate Interference, Dispersed Errors:**: As interference increases, retrieval accuracy drops, and the output distribution spreads. Retrieval errors now stem from much earlier updates—far upstream from the target value rather than adjacent positions (bins). with a small but growing fraction now involve values never presented at all ("hallucinations"). This increasing dispersion marks rising proactive interference and a decline in retrieval fidelity.

**Stage 3 – High Interference, Hallucinatory Responses** At high levels of interference, retrieval accuracy collapses and the model's output distribution undergoes a qualitative shift. The model increasingly returns values that never appeared in the prompt—so-called hallucinations. At the same time, a substantial portion of errors remains anchored to the earliest bins, reflecting a persistent primacy bias toward the first few updates for each key, even as retrieval fidelity breaks down. This change in retrieval behavior resembles a phase transition: once the model's anti-interference capacity is exhausted, it no longer retrieves plausible candidates, consistent with limited-resource theories of working memory failure.

Figure 22 illustrates this progression: as the update count increases (moving left to right in the panels), the model's incorrect responses shift from the most recent value to much earlier, outdated values, and eventually to off-target 'hallucinated' values.

To strengthen the generalizability of our findings, we conducted additional experiments on a broader set of models, with consistent results shown in Supplementary Figure 22

### B.0.2    SIZE OVER INPUT CONTEXT WINDOW

Statistical tests confirm that anti-interference performance correlates with model size and is weakly correlated with the context window.

To quantify each model's robustness to interference, we introduce the Interference Endurance Score (IES). The IES is defined as the area under the curve (AUC) of retrieval accuracy, calculated across log-scaled update counts. We measure how well a model maintains accurate retrieval as interference increases, with a higher IES indicating greater resistance to interference. For comparability and statistical reliability, we compute the IES using the accuracy-versus-update-count function (see Figure 5), which is available for all models tested.

To determine whether model performance is driven more by parameter size or by context length, we conducted a regression analysis of the Interference Endurance Score (IES) against both variables. We grouped models into four parameter size classes—XS, S, M, and L—as defined in Figure 2. Reasoning models were excluded because their more extensive inference processes caused latency to exceed 200 seconds per task, preventing most tests from completing. To minimize noise from closed-source models with uncertain parameter counts, we focused on these defined size classes and restricted our analysis to open models.

The regression (in Table 1 ) shows that parameter size class is a significant predictor of IES (t = 3.03, p = 0.005, N = 30), while context length has no significant effect (t = –0.144, p = 0.886). The combined model explains 26.1% of the variance in IES ($R^2 = 0.261$). To further clarify the role of model size, we performed a separate analysis restricted to models with similar context lengths (128k–131k tokens), which encompasses two-thirds of the non-CoT models. Within this range, the Spearman correlation between parameter size and IES remains strong and significant ($\rho^2 = 0.673$, p = 0.0016; see Figure 8). .

Our analysis shows that

**Model size—not context window length—is the primary factor that underlies robustness to interference.**

**MoE architectures underperform dense models with comparable total parameters** (we conjecture that this is because the number of activated parameters in an MoE model is much smaller than its nominal total).

In cognitive science, performance under proactive interference is a classic probe of working-memory capacity: individuals with greater ability to maintain and manipulate information show greater resistance to interference. Our findings reveal a striking parallel in large language models (LLMs). Across all tested LLMs, we observe a consistent, characteristic decline in retrieval accuracy as inter-

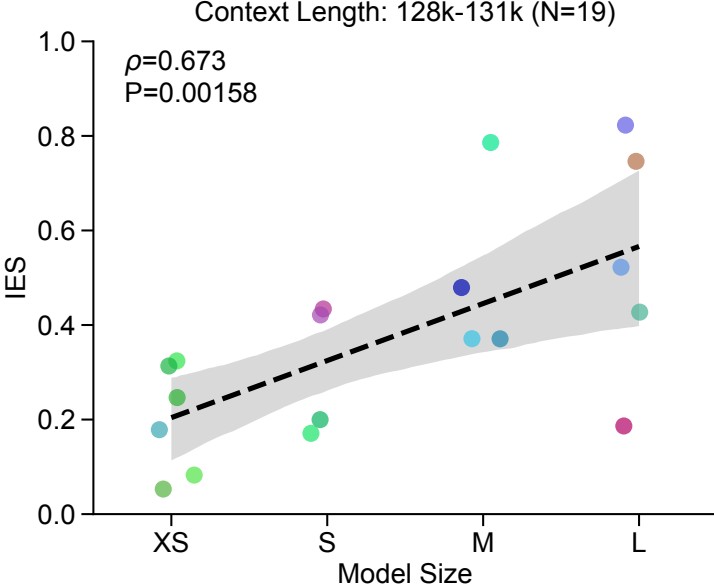

Figure 8: Interference Endurance Score (IES, from Figure 19) shows a strong correlation with model size class (XS, S, M, L; as defined in Figure 2). Each dot represents a model, color-coded as in Figure 19. A linear regression line is included for visualization, with shaded regions indicating 95% confidence intervals. The analysis is restricted to models with similar context lengths (128k–131k tokens, covering about two-thirds of tested non-CoT models. R-squared value is derived from Spearman correlation.

ference increases. Moreover, larger models demonstrate greater resistance to interference—a pattern reminiscent of individual differences in human working memory.

This universal decline, present even in state-of-the-art models spanning a wide range of scales, training data, and computational resources, suggests that limited resistance to interference is an inherent property of transformer-based architectures, rather than a byproduct of specific model size or dataset.

Importantly, our metric captures more than just the context window length or the sheer amount of information a model can store. It quantifies each model's effective ability to manage and control information in the presence of substantial distractors—tracking, updating, and selectively retrieving relevant data amid interference. Thus, anti-interference performance reflects not only storage capacity, but also the executive control processes that underlie working memory in humans. This framework enables us to operationalize and compare the working-memory-like functions of LLMs and human cognition on a principled, quantitative basis.

## C  ROBUSTNESS: PROMPT VARIATIONS AND SEQUENTIAL MODE

Although absolute retrieval accuracy can shift with changes in prompt wording (He et al., 2024), our study emphasizes the relative trend of performance decline rather than raw accuracy scores. This approach effectively neutralizes variability arising from individual prompt formulations.

To further confirm the robustness of the observed proactive interference (PI) effect, we tested additional prompt templates explicitly designed to verify task comprehension. Specifically, we introduced meta-relevant prompts that first ask the LLM to articulate the "task mission"—for example, explicitly prompting the model to "describe the goal of this task" before retrieval. This step ensured the models fully understood the retrieval objective of identifying "the last value" (see Figure 5, 'Relevance meta-prompt'). Across these prompt variations, the qualitative trend of performance decline—specifically, the log-linear decay in accuracy—remained consistently robust.

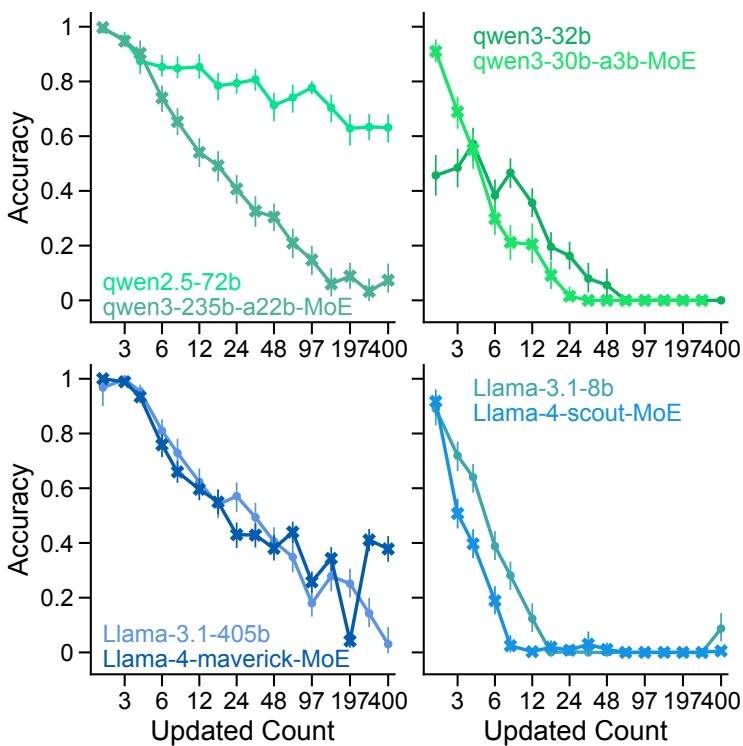

Figure 9: Comparison of retrieval accuracy between Mixture-of-Experts (MoE) and dense models. Each curve shows retrieval accuracy versus update count for a single model. MoE models are denoted by "X" markers and labeled with "MoE" in the legend. Across update counts, MoE architectures consistently match or underperform dense models with similar total parameter counts, and in many cases perform comparably to much smaller dense models. (MoE models shown: Llama-4-maverick-MoE (400B), Llama-4-scout-MoE (109B), Qwen3-30B-A3B-MoE (30B).)

Additionally, we rearranged the input organization to test PI under both randomly shuffled and strictly sequential update sequences (i.e., sequential key–value updates without randomization; see Figure 10). Notably, in sequential mode, retrieval accuracy remains stable until reaching a model-specific interference threshold, after which performance sharply and consistently drops to near-zero—a two-plateau, step-like pattern contrasting with the gradual log-linear decay observed in random mode. The consistent interference-induced decline across diverse models and input structures further underscores the robustness and generalizability of the observed PI phenomenon.

# D   INTERFERENCE IS INDEPENDENT OF INPUT LENGTH

Retrieval accuracy in language models declines log-linearly as the update count per key increases, suggesting a limited working-memory-like capacity. However, in the previous experiment, input length was not controlled; thus, the observed decline might simply reflect increasing context length rather than genuine interference. To directly test the role of interference, we designed two additional Settings.

1. **Number of Updated Keys**: Increasing the number of distinct keys that are updated within the context, while holding the update count per key constant.

2. **Partial Query at Fixed Input Length**: Fixing both the total number of keys and the update count per key (thus keeping the input length constant), but varying the number of keys queried—asking the language model to track and retrieve only a subset of the keys presented.

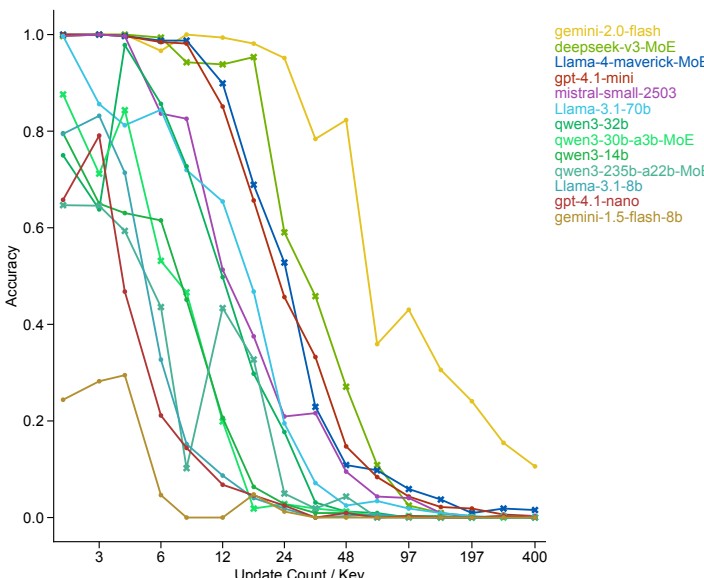

Figure 10: *Step-like failure pattern in sequential key–value update tests.* Retrieval accuracy remains near-perfect as interfering information is added in strictly sequential order, until a model-specific threshold is reached—after which performance drops abruptly to near-zero. Within the same model family, larger models exhibit a higher threshold (better capacity). Despite quantitative differences, all models show the same two-plateau, step-function pattern, reflecting a hard capacity limit. This stands in contrast to the gradual log-linear decay observed under random update order (see Figure 2). (x-axis: number of interfering items, log-scaled; asterisk: MoE models)

By manipulating interference both with and without changes in input length, we can dissociate the effects of interference from those of context length; observing similar declines in retrieval accuracy across both settings would provide strong evidence that interference, rather than context length alone, constrains model performance.

## D.1 EXPERIMENT SETUP

### D.1.1 SETTING A: VARYING THE NUMBER OF UPDATED KEYS

In this experiment, we fixed the update count for each key (either 125 or 350 update count per key), and systematically increased interference by varying the number of distinct keys presented in the sequence-the Updated Keys ($N_U$, from 2 to 46). This contrasts with our earlier experiment, which held the number of keys constant while varying the update count per key.

For each input sequence, there are $N_U$ relevant key–value pairs, with the retrieval target being the last value for each key. Depending on the update count, this results in $N_U \times (125 - 1)$ or $N_U \times (350 - 1)$ irrelevant, interfering key–value pairs per input sequence. Retrieval accuracy was measured as a function of $N_U$-Updated Keys .

### D.1.2 SETTING B: FIXED LENGTH VERSION

To further isolate the effect of interference, we designed a complementary experiment in which the total input length was held constant. In this condition, both the update count per key and the number of updated keys ($N_U$) were fixed, so each input sequence contained the same number of key–value pairs. However, we varied the number of keys the model was instructed to track and retrieve at the end—these are the Tracked Keys ($N_T$), chosen from among the $N_U$ updated keys.

Specifically, the update count for each key was fixed (either 125 or 350 updates per key), and the number of updated keys $N_U$ was also fixed at 46. We then systematically varied the number of

Figure 11: Input example illustrating how the model is prompted to track and return values for a subset of updated keys, as specified by the parameters tracked keys and updated keys. In this minimal example, the tracked keys include "visual art" (blue) and "landform" (orange); the "tools" key (prefixed with a gray index like "1*") appears in the update stream but is not referenced in the initial instruction or final query. Ideally, the model should return only the most recent values for the tracked keys. This setup enables testing whether model performance depends primarily on task-relevant information, rather than irrelevant updates or input length. Bold text highlights the target key-value pairs the model is expected to retrieve.

tracked keys ($N_T$, from 1 to 46), i.e., the subset of keys for which the model was asked to report the final value. Retrieval accuracy was measured as a function of $N_T$, the number of tracked keys.

Figure 11 provides an example input: among $N_U$=3 distinct keys updated in the sequence, only $N_T$=2 are tracked (queried) at the end.

## D.2 RESULTS FOR BOTH SETTINGS

### D.2.1 SETTING A RESULT

Increasing interference by raising the number of updated keys consistently produced a log-linear decline in retrieval performance across all tested model sizes (Left Panel of Figure 12 ). Notably, even though each key received a fixed number of updates—ensuring a constant interference load per key—requiring the model to retrieve the final values for a greater number of keys more rapidly exhausted its anti-interference resources, leading to a substantial reduction in accuracy.

Specifically, in the Left Panel of Figure 12, the x-axis represents the total number of Updated Keys, and models are instructed to track all Updated Keys. Each key's update count is fixed at two values: 125 (upper panel) and 350 (lower panel). The overall trend in log-scale is a linear decline in accuracy, independent of the number of updates per key.

### D.2.2 SETTING B FIX LENGTH RESULT

Retrieval performance exhibits a consistent log-linear decline across all tested models (Right Panel of Figure 12 ). The x-axis represents the total number of Tracked Keys. Notably, larger models show shallower declines than smaller ones, reflecting greater resistance to interference. Under fixed input length, increasing the number of simultaneously tracked keys leads to lower accuracy, in line with this log-linear pattern. For instance, llama4-maverick achieves nearly 100% accuracy when tracking just two keys, but this drops below 5% when tracking 46 keys, consistently following

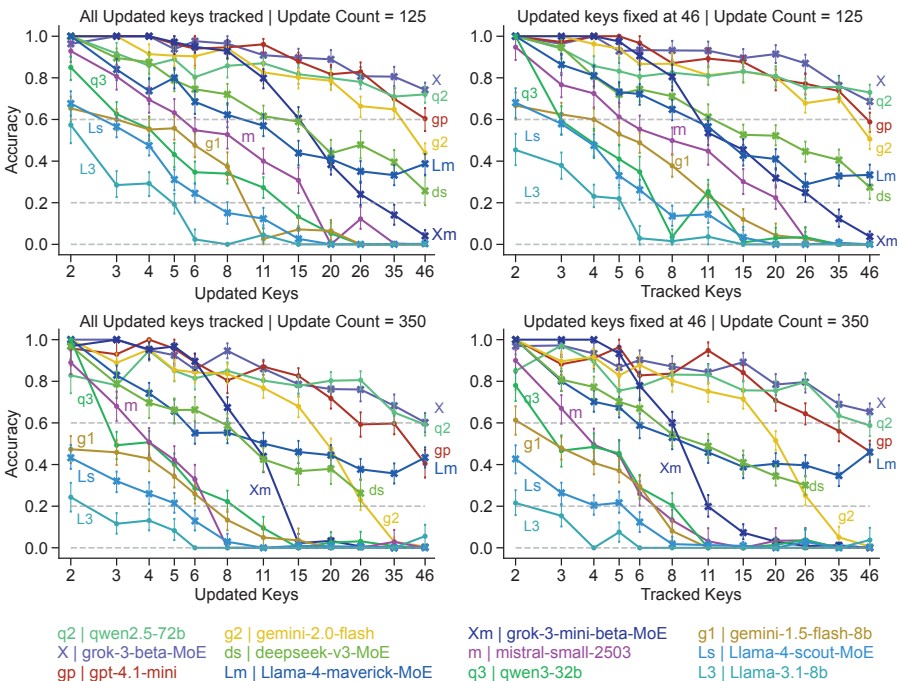

Figure 12: Varying the number of updated keys (left panels) versus the number of tracked keys (right panels, with updated keys fixed at maximum) yields only minor differences in retrieval accuracy. In all conditions, accuracy declines approximately log-linearly with the number of keys. Each key is updated a fixed number of times–125 in the upper panels and 350 in the lower panels. Some models exhibit a two-phase decline; for example, grok-3-mini-beta maintains high performance early on, followed by a sharp drop after a turning point. deepseek-v3 does not complete the full range in the lower panel due to context length limitations. MoE models are indicated by "X" markers. Error bars represent 95% confidence intervals computed via bootstrapping. Model acronyms are used to label the corresponding curves.

the same downward trajectory. These results indicate that, under fixed-length conditions, tracked keys compete for a limited pool of anti-interference resources, which are rapidly depleted as their number grows. Practically, this suggests that reducing the number of concurrently tracked keys can substantially improve retrieval accuracy.

### D.2.3  COMBINED OBSERVATIONS (SETTINGS A AND B)

We observe that both Experiments A and B exhibit nearly identical log-linear declines in retrieval accuracy as the number of tracked keys increases (see Figure 12, left and right panels). Notably, this occurs even though Setting B keeps input length fixed while Setting A allows it to grow. This similarity indicates that the observed performance drop cannot be attributed solely to longer input sequences; rather, it is driven by increased interference from tracking more keys.

Furthermore, models that excel in the variable input length setting (Setting A) also perform well in the fixed-length setting (Setting B), underscoring the robustness of this pattern across experimental setups.

**The universal log-linear decline observed, even under fixed input length, suggests that anti-interference capacity operates as a distinct resource, separate from the total context window length.** In other words, regardless of how much context the model can technically process, its ability to manage interference is independently limited. This distinction highlights that interference resistance is a fundamental capability of LLMs—determined not by context window size, but by deeper architectural or computational constraints within the model.

### D.2.4 DISCUSSION/IMPLICATIONS

These findings have important implications for both model evaluation and practical deployment. They suggest that simply increasing the context window or scaling up input length does not directly translate into better interference management. Instead, targeted advances in anti-interference mechanisms or executive control within model architectures may be needed to substantially improve retrieval accuracy when handling many competing, similar items. This perspective reframes interference resistance as a critical axis of model capability, worthy of focused research and explicit benchmarking alongside traditional context-length and parameter-count metrics.

## E   RETRIEVAL CAPACITY IS LIMITED BY A UNIFIED INTERFERENCE BOTTLENECK ACROSS DIMENSIONS

Value Length = 3

INPUT

As my secretary, I need you to carefully read a text stream where the values of multiple keys are being continuously updated. The 3 keys to track include visual art, tools, landform. I will ask you to identify the current value of each key later. The text stream starts on the next line.
1*visual art: BraqueBasquiatPopart;
1*tools: HookremoverNeedlePaintstic;
2*visual art: PollockCityscapeCubism;
1*landform: PingoMoraineCanyon;
2*tools: PlungerJackstandGlasscutter;
2*landform: PlainBlockfieldScoriacone;
3***tools: RulerHammerScaffolding**;
3***visual art: SeascapeGraffitiCeramics**;
3***landform: DuneValleyHimalayas**.
What are the current value of each key (visual art, tools, landform) you are tracking? End the response with: 'The current value of <key> is <value>.' Provide the exact current value string without modification or breaking it into pieces.

DESIRED ANSWER

The current value of visual art is SeascapeGraffitiCeramics. The current value of tools is RulerHammerScaffolding. The current value of landform is DuneValleyHimalayas.

Figure 13: Input example with manipulation of the updated values's length. In this example, three items from the same category are space-removed, capitalized at the first letter, and concatenated into a single updated value. Bold text indicates the target key-value pairs the model is expected to retrieve.

If an LLM's anti-interference capacity is truly analogous to human working memory, then manipulations that increase working memory demands in humans should produce comparable effects in LLMs. One such manipulation is the classic word-length effect: in human memory research, increasing the length of words to be remembered impairs performance, as longer items consume more working memory resources (Baddeley et al., 1975). This phenomenon provides an additional axis along which working memory capacity can be taxed.

To probe whether LLMs exhibit a similar sensitivity, we systematically varied the length of words within key–value pairs by concatenating multiple words into each value. This allowed us to directly test whether increasing the information load per item would similarly degrade retrieval performance in LLMs.

In this experiment, we held constant the three previously identified sources of interference: the number of updates per key, the number of updated keys, and the number of keys to track. To manipulate interference strength in line with the classic word-length effect observed in human working memory, we systematically increased the length of the updated value strings. Specifically, we concatenated multiple dictionary words end-to-end (e.g., **AppleOrangeBanana**), thereby **increasing both the word length and the token count—the fundamental unit of LLM processing.** This manipulation closely mirrors the increased cognitive load humans experience when encoding longer words in memory tasks. Figure 13 in the Appendix provides an example input.

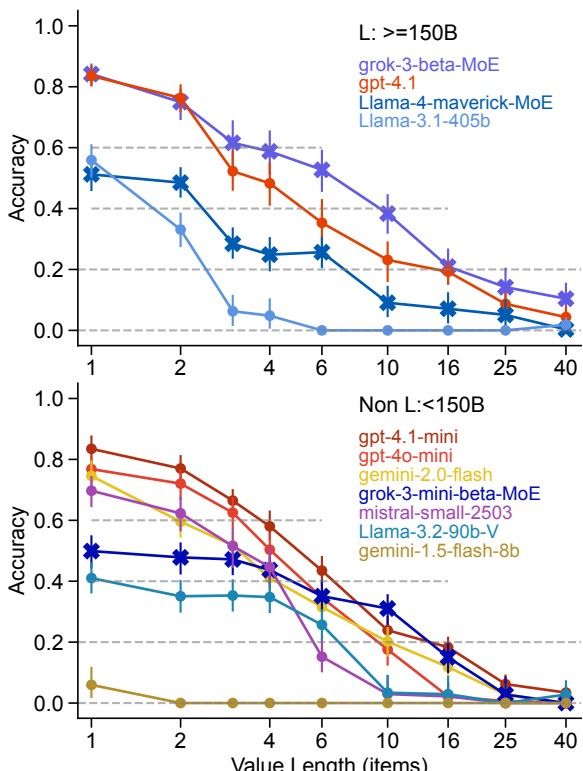

Figure 14: Retrieval accuracy as a function of value length, showing a roughly log-linear decline toward near-zero performance. For clarity, models are grouped by parameter size: large models (L; ≥150B parameters) are shown in the upper panel, and smaller models (< 150B) in the lower panel. The update count is fixed at 20. Some models exhibit an initial plateau phase, with stable accuracy for short value lengths (ranging from 1 to 4). At the largest value length tested, accuracy drops to near zero for most models, with the exception of Grok-3-beta, which retains a performance of approximately 0.1. MoE models are indicated with "X" markers. Error bars represent bootstrapped 95% confidence intervals.

### E.1 RESULTS AND INTERPRETATION

**LLMs exhibit a universal, approximately log-linear decline in retrieval accuracy as the length of each value increases.** The slope of this decline is markedly steeper than for the other three interference manipulations: increasing value length from one to ten words drives accuracy below 40% for every model tested, and extending it to forty words reduces accuracy to under 5%. Notably, this sharp drop occurs even when the number of keys, updates, and tracked keys is held constant, highlighting the unique impact of item length.

This result demonstrates that increasing the amount of information stored in each retrieved value—by concatenating more words—adds a distinct, independent dimension of interference, taxing the system's capacity beyond what can be explained by the number of tracked keys or updates alone. The effect of value length thus exposes another axis along which the model's anti-interference resource can be depleted.

This outcome closely parallels human memory performance, where recalling longer or more complex words substantially lowers accuracy—a classic word-length effect. Taken together with prior results, these findings reinforce our explanatory framework: all forms of interference—whether from more keys, more updates, or longer values—tap into a single, **unified anti-interference resource in the model, analogous to a working-memory buffer.** As the informational load per item grows, this capacity is consumed more rapidly, leading to steeper performance degradation. This unified capacity constraint, shared across all tested dimensions, underscores a structural limitation in current LLM architectures that mirrors properties of human working memory.

## F MITIGATING INTERFERENCE: EMPIRICAL INSIGHTS FROM LLM–HUMAN COMPARISON

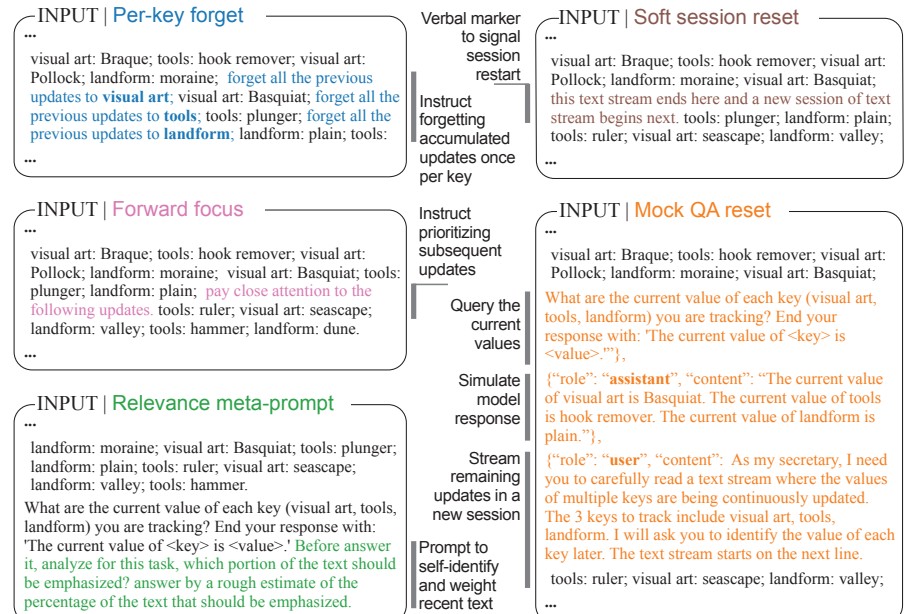

Figure 15: Example input illustrating intervention strategies designed to mitigate proactive interference. Each strategy inserts explicit cues into the update stream, typically near the end (e.g., at the 120th-last update or one-third before the final update). The five strategies are: *Per-key forget* (green): An instruction to disregard previous updates for a specific key before a new one (e.g., "forget all the previous updates to visual art"). *Forward focus* (magenta): An instruction to prioritize information that follows (e.g., "pay close attention to the following updates"). *Relevance meta-prompt* (green): A prompt for the model to self-assess and estimate the proportion of text to prioritize before answering. *Soft session reset* (brown): A verbal cue marking the start of a new input segment (e.g., "this text stream ends here and a new session of text stream begins next"). *Mock QA reset* (orange): A simulated dialogue turn including an initial update segment, a query, a mock assistant response, and remaining updates in a new user turn. Inserted instructional cues are shown in colored text; role labels in the Mock QA reset are in bold.

Our previous experiments demonstrated that LLMs possess limited anti-interference capacity, with retrieval accuracy declining log-linearly as interference increases. To better understand and potentially mitigate this limitation, we compared LLM performance to humans on the same key-value retrieval task, drawing on strategies from human cognitive experiments to design corresponding interventions for LLMs.

In contrast to LLMs, humans exhibit a plateau in recall accuracy for the most recent key-value pair, even as the number of prior updates accumulates. Classic working memory studies attribute this resilience to executive control mechanisms. Two particularly relevant mechanisms are gating, which automatically suppresses or discards outdated information as new items are encoded (Oberauer & Vockenberg, 2009), and directed forgetting, where individuals intentionally discard certain information when explicitly instructed to do so (Festini & Reuter-Lorenz, 2014).

LLM retrieval accuracy declines continuously with increasing interference, suggesting the absence of automatic gating mechanisms. Moreover, humans can engage in explicit, strategic forgetting. To test whether LLMs might benefit from such strategic forgetting, we simulated this human capability by providing LLMs explicit prompts instructing them to forget previous key-value pairs. If successful, such an intervention would demonstrate that LLMs can emulate human-like release from interference through external cues, potentially alleviating the limitations of their anti-interference capacity.

### F.1 Simulating Human Directed Forgetting with Natural Language Prompt

To simulate the human strategy of explicit directed forgetting, we inserted a targeted prompt into the input sequence that directly instructs the LLM to disregard all prior updates for a specific key. This prompt is placed at a fixed point in the update stream—immediately before the chunk containing the majority of the target answer, after most interfering updates have been presented. The directive reads: "Forget all the previous updates to key," with key dynamically replaced by the relevant key for the current task.

The purpose of this intervention is to actively suppress the influence of outdated or distracting information from earlier in the sequence, thereby reducing proactive interference and guiding the model to prioritize only the most recent updates for retrieval. This approach tests whether an explicit natural language cue can shift the model's focus in a way that mimics human executive control over memory. See Figure 15 for examples of the "Per-Key Forget" prompt.

For comparison, we also tested a "Forward Focus" prompt, which instructs the LLM to concentrate on the more recent, relevant portion of the input. This allows us to evaluate whether explicit natural language instructions—whether aimed at forgetting or focusing—can meaningfully affect model retrieval performance.

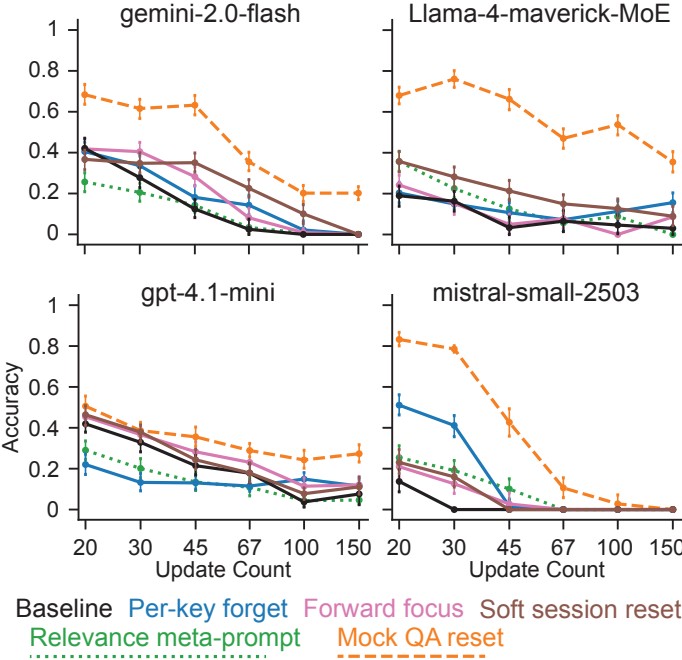

Figure 16: Explicit forgetting and focusing prompts inserted during the update stream (as shown in Figure 15 yielded only marginal improvements in retrieval accuracy. The black line indicates the baseline condition with no intervention prompt. Solid lines represent several simple natural language prompts designed to instruct the model to forget previous updates, focus on upcoming ones, or reset context. For most models, these interventions had limited effect, especially at higher update counts, where the baseline performance is low. The per-key forget even had a negative effect on gpt-4.1-mini. The relevance meta-prompt (green dotted), which asked the model to self-assess what to focus on, was ineffective for all models and even harmful for gpt-4.1-mini. Only the mock QA reset intervention (orange dashed line), which simulates a user-model interaction, led to a substantial improvement in retrieval accuracy. However, this strategy was not immune to the overall trend: accuracy continues to decline with increasing update count (log-spaced).

## F.2 NATURAL LANGUAGE PROMPT FAILS

The per-key forget prompt—designed to mimic human explicit forgetting—did not significantly improve retrieval accuracy (blue line in Figure 16; $\Delta < 10$ percentage points compared to the baseline at 100 updates, black line). Similarly, alternative natural language instructions intended to focus the model on the target retrieval section were also ineffective. Overall, **natural language instructions—whether to forget or focus—do not effectively reduce interference in LLM retrieval tasks.**

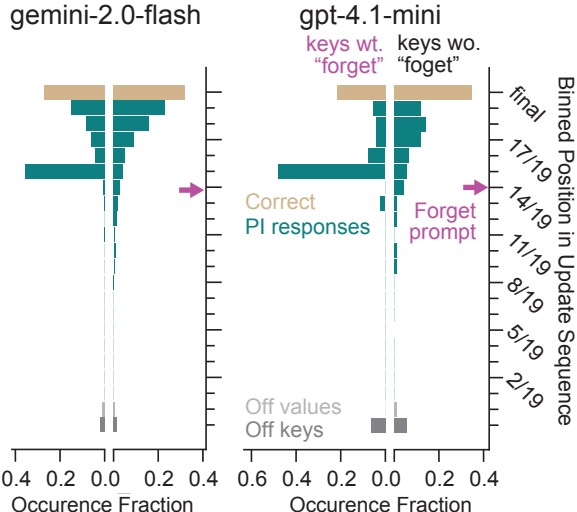

Figure 17: A selective per-key forgetting prompt induces a distinct pattern of Proactive Interference (PI): instead of enabling successful forgetting, the prompt causes retrieval errors to cluster around the position in the update sequence where the instruction was injected. The figure compares keys that received a forgetting instruction with a control group; only the instructed keys show this pronounced error concentration, indicating that the prompt anchored the model's retrieval errors to that part of the sequence rather than erasing the information. Earthy yellow bars indicate the correct value—the final update. Green bars represent earlier (interfering) values, grouped by their relative position in the update sequence. Light gray bars show "off values" not present in the update history, and dark gray bars denote "off keys," where the model failed to return any value. The results shown are from an experiment with 20 updates per key and 46 unique keys. For a comprehensive analysis across various model architectures and a wider range of parameter settings, see Figure 23 in the Appendix.

An analysis of the error distribution reveals a critical failure mode: rather than improving retrieval accuracy, the per-key forgetting prompt consistently caused errors to cluster around the position in the sequence where the instruction was injected. As shown in Figure 17, models displayed a pronounced tendency to select earlier values immediately preceding the forget instruction, rather than the correct, final update. This error pattern indicates that the prompt did not enable the model to effectively disregard prior information. Instead, it induced a concentration of retrieval errors near the instructed forget position, reshaping interference rather than mitigating it. In summary, **rather than mitigating interference, these prompts cause errors to cluster around the location of the prompt, indicating that the model's anti-interference limitation cannot be overcome by simple natural language cues.**

## F.3 HACK METHOD: MOCK-QA-RESET SUCCEEDS

Inspired by LLM "hacking" studies showing that models can be coaxed to bypass earlier instructions (Kuo et al., 2025), we devised a non-natural-language mock QA reset prompt (see Figure 15) that mimics human directed-forgetting. Inserted 120 updates before the final query, this reset cue leads the model to treat preceding input as belonging to an already processed prior task, thereby partially mitigating interference and improving retrieval accuracy. While this ad hoc prompt inter-

vention partially reduces interference, it highlights the need for more systematic methods to address interference in LLMs' retrieval task.

The prompt has three parts:

- **Simulated user query** asking for the current value of all tracked keys (e.g., "User: What is the current value of key1, key2, ..., key45?"), which frames prior updates as a closed batch.

- **Simulated assistant reply** giving fabricated answers (e.g., "Assistant: The current value of key1 is ..., key2 is ..."), providing explicit closure.

- **New user prompt** signalling a fresh tracking task, followed by the remaining updates (e.g., "User: I will now provide 45 updated key–value pairs. Tell me the most current value. The pairs begin:"), which marks a clear task boundary and encourages the model to ignore earlier content.

This artificial task boundary partially mitigates interference by prompting the model to deprioritize earlier input and focus on newly updated information.

**The hack prompt substantially improved retrieval accuracy (As shown in Figure 16, orange line), reducing the effects of interference across all tested LLMs.** This hack-based prompt consistently outperformed natural language instructions designed to induce forgetting or refocusing. For example, with the hack-reset, Gemini Flash 2.0's retrieval accuracy at 150 update count—under high interference—matched its baseline performance at only 30–45 update count, demonstrating a substantial reduction in interference effects.

The success of our hacking-based reset method demonstrates that implementing a gating mechanism can effectively reduce interference in LLMs, closely mirroring the executive gating functions of human working memory. This result suggests that implementing gating mechanisms in LLMs could be an effective strategy for reducing interference, mirroring the executive functions of human working memory.

However, while our reset strategy shows that LLMs benefit from artificially imposed context boundaries, this approach remains fundamentally limited. Specifically, our intervention mitigates interference by diminishing the influence of all prior information—effectively discarding or bypassing past associations. Although this provides short-term relief, it is not a viable solution for real-world tasks, which frequently require selective, context-dependent access to historical data beyond just the most recent update.

This limitation is further highlighted by additional experiments with natural-language prompts designed to instruct LLMs to ignore or forget prior information—such as the 'soft session reset' shown in Figure 15—which were largely ineffective (see performance in Figure 16). These findings indicate that current LLMs cannot be reliably controlled through explicit natural-language user instructions alone; precise, natural-language-based adjustments of memory and attention remain an open challenge.

## G SUMMARY

Our systematic investigation of proactive interference (PI) in Large Language Models (LLMs) across various scales from 0.6B to over 600B, reveals a pervasive susceptibility to interference effects during retrieval tasks. Critically, LLMs demonstrate a continuous log-linear decline in retrieval accuracy as interference increases, showing no evidence of a plateau. Moreover, the continuous decline—characterized by a similar log-linear pattern—emerges independently along multiple dimensions of interference load: the number of sequential updates to a key, the number of keys tracked concurrently, and the token length of each updated value. The convergence of these qualitatively similar decline patterns, across orthogonal axes of load, implies that LLMs possess a finite, resource-like representational capacity that can be incrementally taxed by different, yet functionally interchangeable, forms of cognitive load, independent of total input size or the model's maximum context length. Collectively, these findings indicate that the anti-interference capacity observed in LLMs closely parallels the properties of human working memory.

We also identify a critical dissociation between the analytical and execution capabilities of LLMs: even models capable of explicitly articulating effective retrieval strategies fail to translate this analytical understanding into improved retrieval performance, underscoring a lack of top-down executive control over retrieval tasks.

Our findings establish proactive interference as a pervasive failure mode in contemporary LLMs and introduce a novel interpretation: a model's resistance to proactive interference directly reflects its underlying working-memory capacity. Unlike traditional metrics that emphasize total input length, our approach reveals interference resilience as a distinct, cognitively-grounded dimension of model capability. Since interference is inherent to tasks ranging from summarizing repeatedly updated information to conducting complex, long-horizon reasoning, enhancing LLMs' working-memory robustness becomes critical for practical performance. By providing a structured synthetic evaluation framework explicitly designed to measure susceptibility to interference, this study offers both a diagnostic tool and a theoretical advance toward understanding and improving LLM cognition. Our code and datasets are publicly released to foster further investigation into the memory mechanisms of large language models.

# H  DISCUSSION

## H.1  METHODOLOGICAL CONTRIBUTION

### H.1.1  FROM COGNITIVE PARADIGMS TO LLM DIAGNOSTIC TOOLS

Our current work aligns squarely with this research thrust. We do not merely suggest that Proactive Interference (PI) is a problem for LLMs by analogy to humans; we adapt the specific experimental logic of the A-B, A-C, A-D paired-associate learning paradigm—a workhorse of human PI research—to create a novel, synthetic diagnostic tool for LLMs. This allows for controlled experimentation and the systematic manipulation of variables, moving beyond correlational observations from general benchmarks towards a more causal understanding of LLM failure modes.

Crucially, the A-B, A-C (and, by extension, A-D, A-E. . . ) schema captures a vast class of real-world problems: streaming sensor readouts, mutable legal ledgers, and long reasoning chains in which the same variable is updated and queried repeatedly. By embedding this ubiquitous "value-overwriting" structure into our testbed, we ensure that the experiment speaks to both practical performance gaps and deeper theoretical questions about how LLMs process interfering information.

## H.2  THEORETICAL EXPLANATIONS AND IMPLICATIONS

Our results suggest that current LLMs possess only an implicit, resource-bounded form of memory selectivity. Self-attention weights provide a quasi-executive filter that suffices for low–interference conditions, but unlike human prefrontal gating, it cannot be strengthened or re-allocated on demand. When the interference budget is exceeded, the model's retrieval accuracy degrades monotonically to near-zero performance, revealing the absence of a true top-down control system. Because adaptive executive control over memory is widely held to be a core component of goal-directed intelligence, these findings point to a critical gap between contemporary LLMs and human cognition: transformers can store vast contexts, but they cannot decide how to use—or forget—them.

### H.2.1  CONNECTING BEHAVIORAL EVIDENCE WITH MECHANISTIC INTERPRETABILITY

Our research also complements work on the mechanistic interpretability of LLMs, such as the study of induction heads (Anthropic, 2022). While induction heads offer a plausible mechanism for how in-context learning and subsequent interference might occur (e.g., an induction circuit strongly encoding A-B might resist an A-C update), our paper provides the experimental paradigm to test the behavioral consequences of such mechanisms when they are confronted with conflicting associative information. Our synthetic setup is designed precisely to probe the conditions under which these induction-like mechanisms are robust versus when they are susceptible to PI.

Recent studies have applied human working-memory (WM) tests, such as the N-back paradigm, to assess the possible WM capacity of LLMs (Gong et al., 2024). While prior work has primarily focused on transplanting classic cognitive tests from human studies to LLMs, our approach integrates

cognitive science methodologies with tasks modeled on realistic LLM applications. This design enables more ecologically valid assessments—reflecting typical model usage—and allows for direct comparison of LLM and human retrieval abilities on matched tasks. Utilizing proactive interference as a framework, we identify specific behavioral differences and practical limitations that standard benchmarks often fail to reveal. These results underscore the importance of integrating cognitive and applied perspectives to advance research on LLM capabilities.

# I   DETAILED GRAPHS AND TABLES

This section presents detailed test figures that complement the main results. We expand the analyses to additional models and variants beyond the main essay, providing per-model summaries, rankings, and distributional diagnostics.

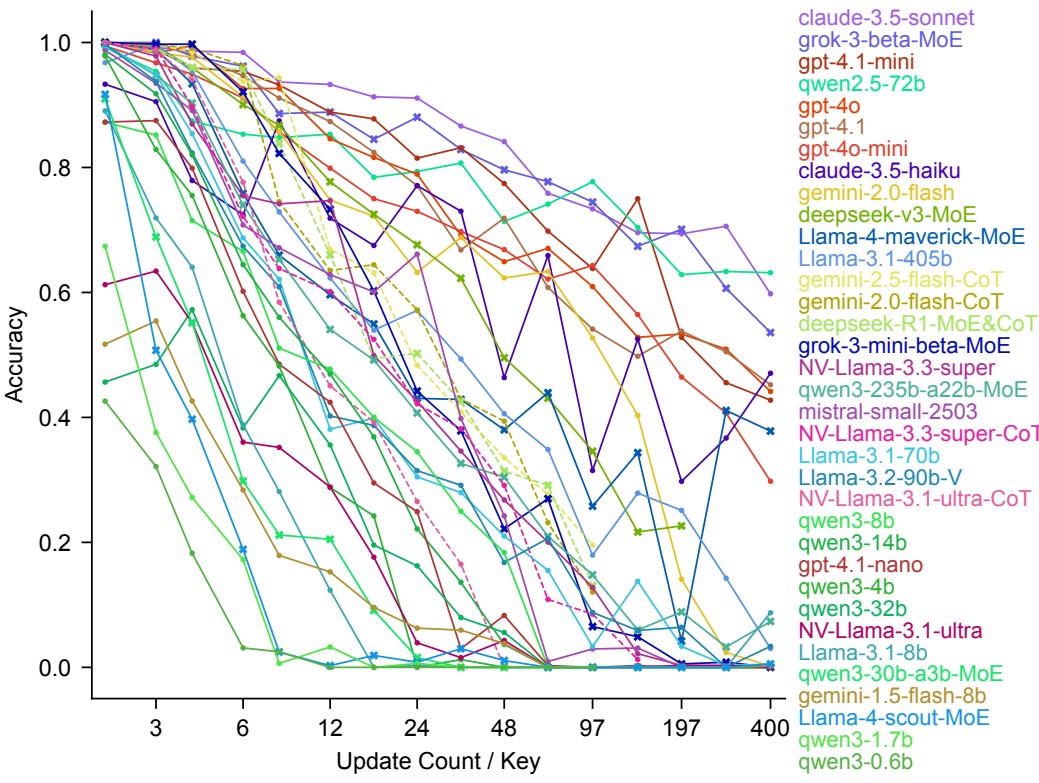

Figure 18: Universal *log-linear* decline in retrieval performance due to interference. Increasing the amount of interfering information preceding a retrieval target within a language model's input context results in a log-linear decrease in retrieval accuracy across diverse models. The target is positioned after the interfering information and explicitly referenced in the prompt to reduce search difficulty and isolate interference effects. (x-axis: the number of Co-referenced information, log-scaled; asterisk: MoE models).

## I.1   MODEL VERSIONS

This section provides a comprehensive list of all language models used in our evaluation. All experiments were conducted up to May 5th, 2025. Models with explicit date stamps in their identifiers (e.g., `gpt-4o-2024-11-20`) represent fixed snapshots. For other models, we used the versions detailed below. The -thinking suffix indicates the model was evaluated with its native reasoning/Chain-of-Thought (CoT) mode enabled; the counterpart without the suffix was evaluated with this mode disabled.

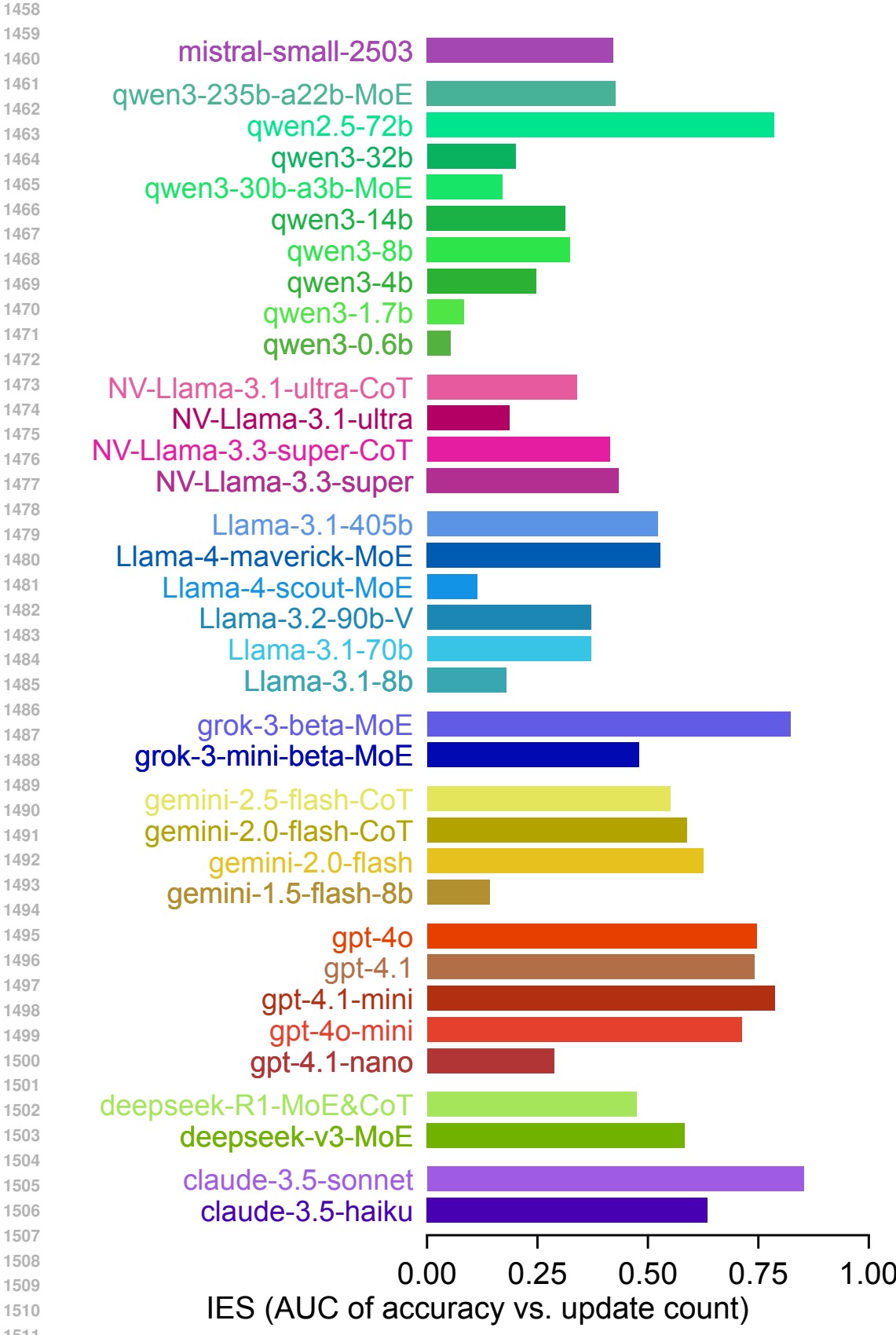

Figure 19: Interference Endurance Score (IES) for all models shown in Figure 3, computed as the area under the curve (AUC) of their accuracy–update-count functions in Figure 2. Higher IES indicates greater robustness to interference across increasing update counts. Models are grouped by family using the same color scheme as in Figure 3, and within each family, sorted by parameter size from large (top) to small (bottom). For a ranking of IES values by magnitude, see Figure 20 in the Appendix

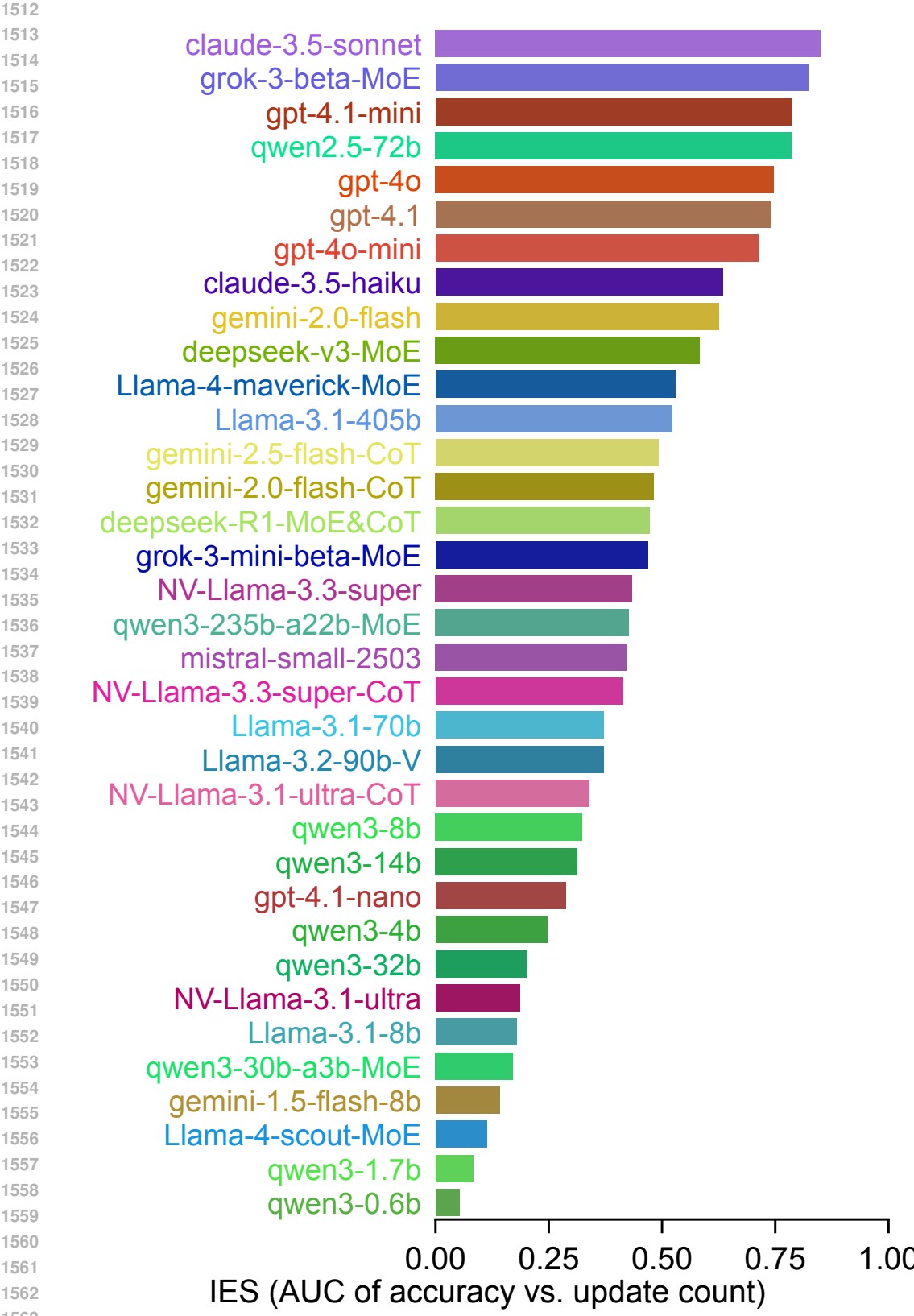

Figure 20: Interference Endurance Scores (IES) from Figure 19, re-ordered by IES value in descending order.

| Variable | Coef. | t | P> \|t\| | [0.025 | 0.975] |
|---|---|---|---|---|---|
| Intercept | 0.1866 | 2.045 | 0.051 | -0.001 | 0.374 |
| Parameter Size | 0.1030 | 3.016 | 0.006 | 0.033 | 0.173 |
| Context Size | -3.334e-09 | -0.144 | 0.887 | -5.08e-08 | 4.41e-08 |

Table 1: Linear Regression Results of Interference Endurance Score (IES) on model parameter size (ordinal class) and model context window. This analysis aims to determine whether model performance is driven more by parameter size or by context length. 30 models were grouped into four parameter size classes (XS, S, M, L)

*Notes.* (1) All experiments were conducted up to May 5, 2025. Models with explicit date stamps in their identifiers (e.g., `gpt-4o-2024-11-20`) represent fixed snapshots, while other identifiers represent the latest available API endpoints as of the cutoff date. (2) The -thinking suffix indicates the model was evaluated with its native reasoning/Chain-of-Thought (CoT) mode enabled; the counterpart without the suffix was evaluated with this mode disabled.

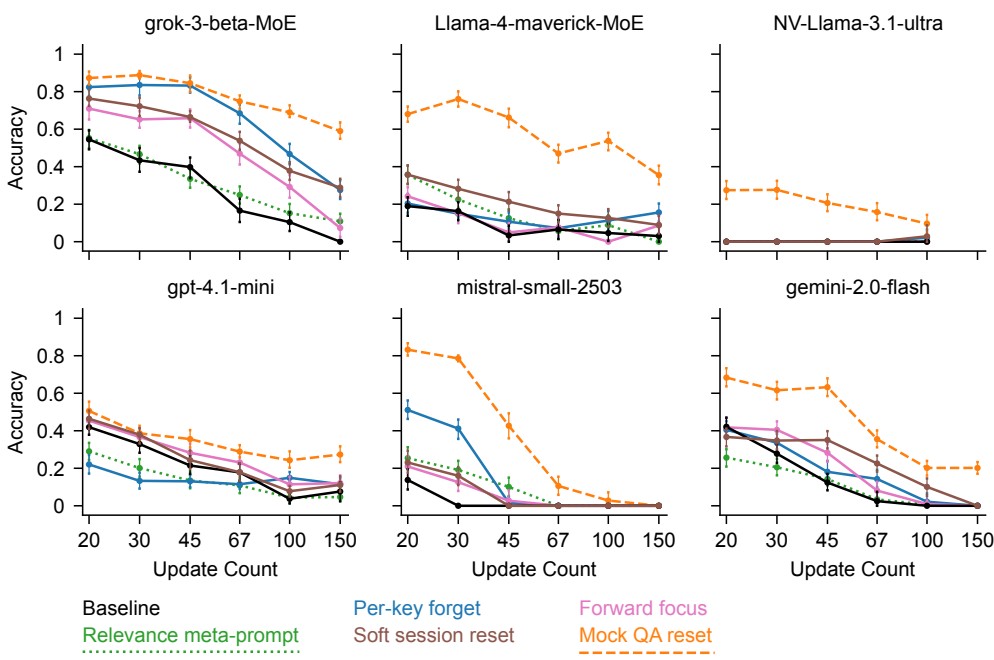

Figure 21: Explicit forgetting and focusing prompts inserted during the update stream (as shown in Figure 15 yielded only marginal improvements in retrieval accuracy. The black line indicates the baseline condition with no intervention prompt. Solid lines represent several simple natural language prompts designed to instruct the model to forget previous updates, focus on upcoming ones, or reset context. For most models, these interventions had limited effect, especially at higher update counts, where the baseline performance is low. The per-key forget(blue line) even had a negative effect on gpt-4.1-mini. The relevance meta-prompt (green dotted), which asked the model to self-assess what to focus on, was ineffective for all models and even harmful for gpt-4.1-mini. Only the mock QA reset intervention (orange dashed line), which simulates a user-model interaction, led to a substantial improvement in retrieval accuracy. However, this strategy was not immune to the overall trend: accuracy continues to decline with increasing update count (log-spaced).Experiments used 46 unique keys and a key-value pair length of 6.

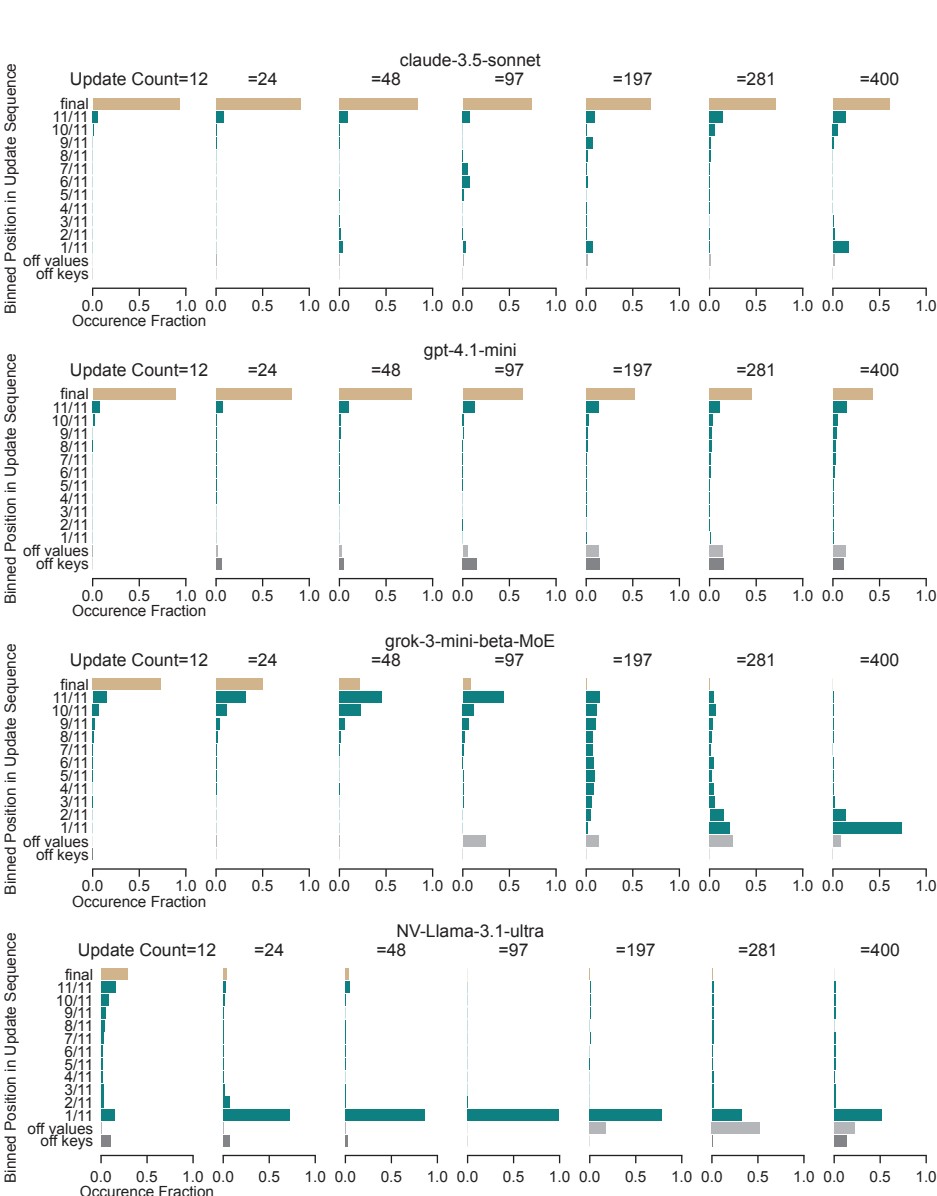

Figure 22: Distribution of model responses across update positions, showing increasing signs of PI as update count increases (left to right). The y-axis lists 11 equal-width bins (Bin 1–Bin 11, green) covering the entire update sequence. The earthy yellow bar indicates the single final update—the correct retrieval target. Light gray bars ("off values") denote cases where the model returns a value not present in the update history (i.e., hallucinations). Dark gray bars ("off keys") indicate failures to return any value for the queried key. As update count increases, errors shift from clustering near the final update to earlier bins, with rising rates of off-values and off-keys.

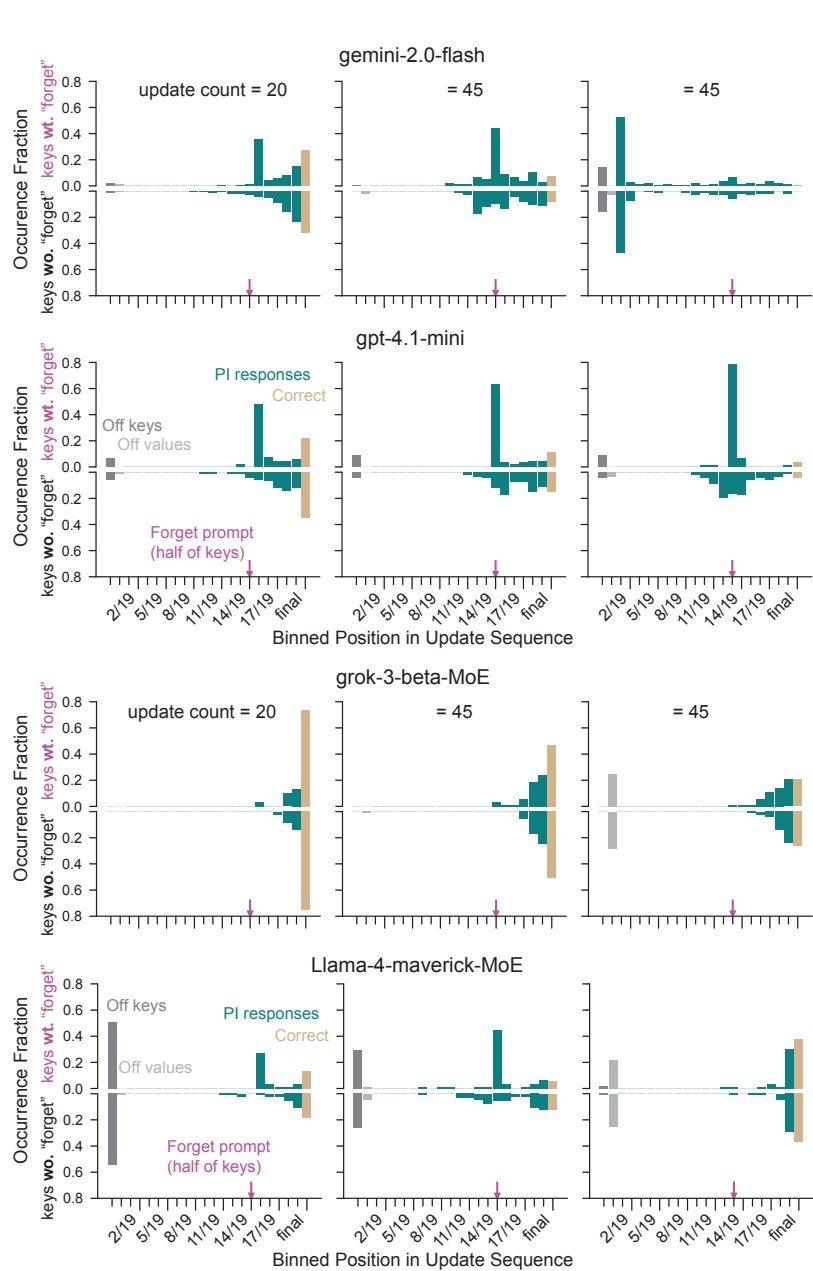

Figure 23: The selective per-key forgetting prompt amplifies proactive interference. Keys that received a forgetting instruction prior to the final third of their updates exhibited concentrated errors around the forgetting point, compared to keys without such a prompt. The x-axis indicates the position of the selected value within the update sequence, categorized for each key. Earthy yellow bars indicate the correct value—the final update. Green bars represent earlier (interfering) values, grouped into 19 bins based on their relative position in the update sequence. Light gray bars indicate "off values" not present in the update history. Dark gray bars denote "off keys," where the model failed to return any value.

| Model Name | Version/Snapshot |
|---|---|
| *Google Gemini Models* | |
| gemini-2.5-flash-preview | gemini-2.5-flash-preview-04-17 |
| gemini-2.0-flash | gemini-2.0-flash |
| gemini-1.5-flash-8b | gemini-1.5-flash-8b |
| gemini-2.0-flash-thinking-exp[2] | gemini-2.0-flash-thinking-exp-01-21 |
| *OpenAI Models* | |
| gpt-4.1 | gpt-4.1-2025-04-14 |
| gpt-4.1-mini | gpt-4.1-mini-2025-04-14 |
| gpt-4.1-nano | gpt-4.1-nano-2025-04-14 |
| gpt-4o | gpt-4o-2024-11-20 |
| gpt-4o-mini | gpt-4o-mini-2024-07-18 |
| *Anthropic Claude Models* | |
| claude-3-5-sonnet | claude-3-5-sonnet-20241022 |
| claude-3-5-haiku | claude-3-5-haiku-20241022 |
| *Alibaba Qwen Models* | |
| qwen2.5-72b-instruct | qwen2.5-72b-instruct[1] |
| qwen3-0.6b | qwen3-0.6b[1] |
| qwen3-1.7b | qwen3-1.7b[1] |
| qwen3-4b | qwen3-4b[1] |
| qwen3-8b | qwen3-8b[1] |
| qwen3-14b | qwen3-14b[1] |
| qwen3-32b | qwen3-32b[1] |
| qwen3-30b-a3b | qwen3-30b-a3b[1] |
| qwen3-235b-a22b | qwen3-235b-a22b[1] |
| qwen3-0.6b-thinking[2] | qwen3-0.6b-thinking[1] |
| qwen3-1.7b-thinking[2] | qwen3-1.7b-thinking[1] |
| qwen3-4b-thinking[2] | qwen3-4b-thinking[1] |
| qwen3-8b-thinking[2] | qwen3-8b-thinking[1] |
| qwen3-14b-thinking[2] | qwen3-14b-thinking[1] |
| qwen3-32b-thinking[2] | qwen3-32b-thinking[1] |
| qwen3-30b-a3b-thinking[2] | qwen3-30b-a3b-thinking[1] |
| qwen3-235b-a22b-thinking[2] | qwen3-235b-a22b-thinking[1] |
| *Meta LLaMA Models* | |
| llama-4-maverick-17b-128e-instruct-maas | llama-4-maverick-17b-128e-instruct-maas[1] |
| llama-4-scout-17b-16e-instruct-maas | llama-4-scout-17b-16e-instruct-maas[1] |
| llama-3.1-405b-instruct-maas | llama-3.1-405b-instruct-maas[1] |
| llama-3.2-90b-vision-instruct-maas | llama-3.2-90b-vision-instruct-maas[1] |
| llama-3.1-70b-instruct-maas | llama-3.1-70b-instruct-maas[1] |
| llama-3.1-8b-instruct-maas | llama-3.1-8b-instruct-maas[1] |
| *DeepSeek Models* | |
| deepseek-chat | deepseek-chat[1] |
| deepseek-reasoner | deepseek-reasoner[1] |
| *xAI Grok Models* | |
| grok-3-beta | grok-3-beta[1] |
| grok-3-mini-beta | grok-3-mini-beta[1] |
| *Mistral Models* | |
| mistral-small-2503 | mistral-small-2503[1] |
| *NVIDIA Models* | |
| nvidia_llama-3.1-nemotron-ultra-253b-v1 | nvidia_llama-3.1-nemotron-ultra-253b-v1[1] |
| nvidia_llama-3.3-nemotron-super-49b-v1 | nvidia_llama-3.3-nemotron-super-49b-v1[1] |
| nvidia_llama-3.1-nemotron-nano-8b-v1 | nvidia_llama-3.1-nemotron-nano-8b-v1[1] |

Table 2: Model versions used in our evaluation

*Notes.* (1) All experiments were conducted up to May 5, 2025. Models with explicit date stamps in their identifiers (e.g., `gpt-4o-2024-11-20`) represent fixed snapshots, while other identifiers represent the latest available API endpoints as of the cutoff date. (2) The -thinking suffix indicates the model was evaluated with its native reasoning/Chain-of-Thought (CoT) mode enabled; the counterpart without the suffix was evaluated with this mode disabled.

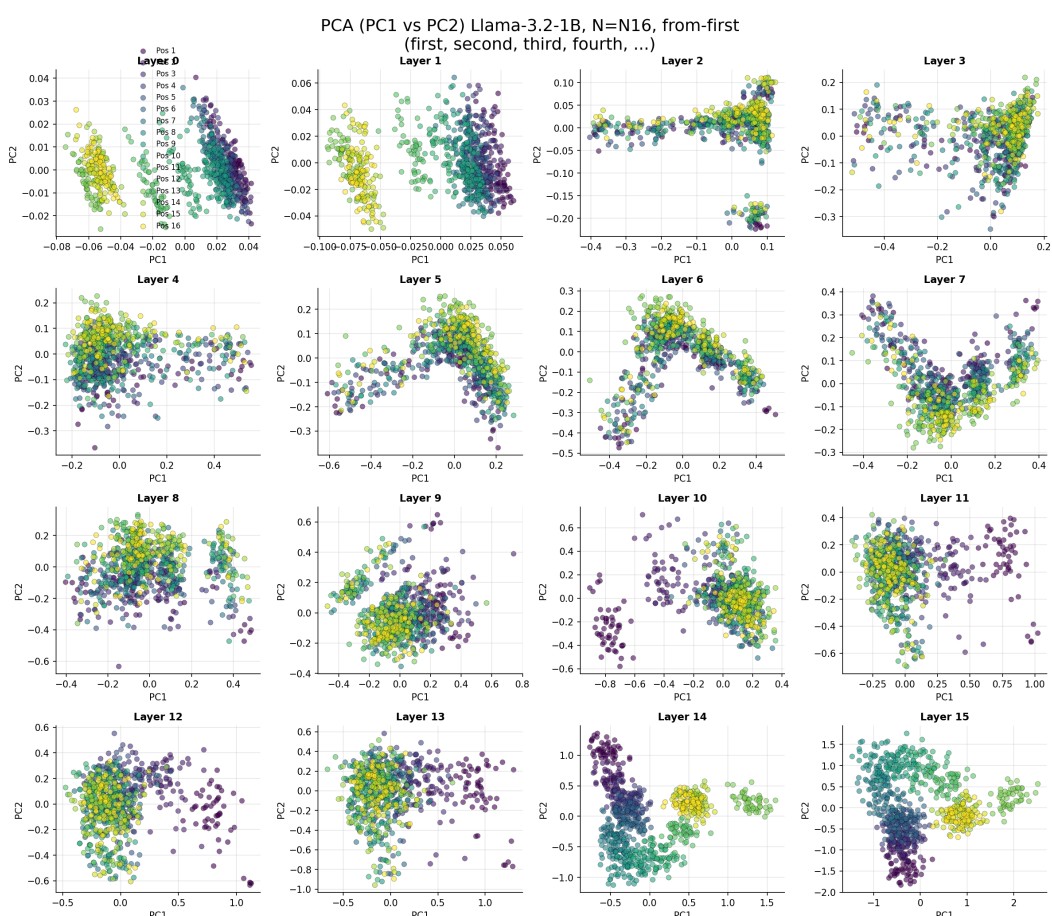

Figure 24: Cross-layer PCA of binding representations in Llama-3.2-1B. We construct a single PI-LLM context where one key (key1) is sequentially updated to 16 distinct values ($N = 16$). For each disambiguating query of the form "What is the $n$-th value of key1?" ($n = 1, \ldots, 16$), we extract the residual-stream activation at the last input token (just before generation) at every transformer layer. Within each layer, we run PCA over the 16 query representations and plot the first two principal components. Each point in a layer's PCA plot corresponds to one query, colored by which value index it targets (e.g., purple = 1st value of key1). Across depth, different value indices initially occupy well-separated regions but become progressively compressed and partially overlapping in later layers, indicating that multiple bindings of the same key are packed into a shared low-dimensional subspace rather than stored as independent, non-interacting slots.

| Submission | Reasoning tokens | Deliberation time | Correct / Total | Accuracy |
|---|---|---|---|---|
| Medium effort | 15.9k | 6 min 25 s | 18 / 46 | 39.1% |
| High effort | 39.9k | 10 min 00 s | 33 / 46 | **71.7%** |

Table 3: GPT-5 proactive-interference (PI) recall performance under medium- and high-effort reasoning (CoT) settings. Even in the high-effort setting, where GPT-5 produces ≈39.9k CoT tokens with latency up to 10 minutes, accuracy reaches only 71.7% (33/46), up from 39.1% (18/46) at medium effort. This shows that stronger SOTA models and more aggressive reasoning settings partially mitigate interference but do not eliminate it: substantial interference-driven failures remain, and performance does not trend cleanly toward ceiling even for GPT-5.

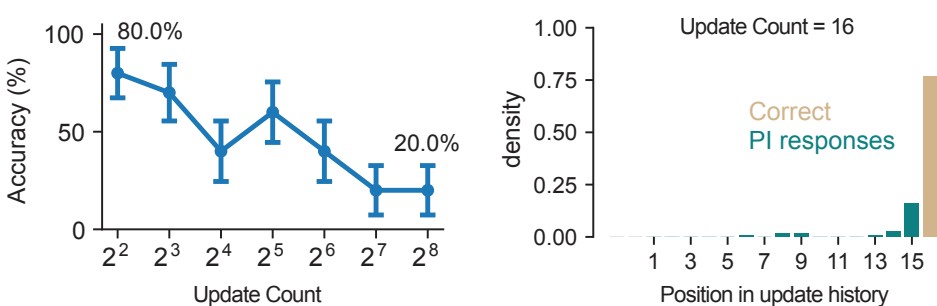

Figure 25: Validation in a naturalistic setting. (Left) Retrieval accuracy of `gpt-4.1-nano` on the Product Launch State Tracking task (multi-turn meeting transcripts). Accuracy shows an approximately log-linear decline as the number of updates increases from 4 to 256, indicating that interference effects persist in naturalistic contexts. (Right) Distribution of retrieved values for an update count of 16. The beige bar marks correct recall of the most recent update; teal bars show incorrect responses. Errors are concentrated on recent but outdated updates (proactive interference), rather than random values from the distant past.

--- START OF SLACK HISTORY EXPORT ---

Channel: #project-alpha-launch

Topic: Launch Event Logistics & Planning

[System]: Project initialized. Waiting for decisions.

[Day 1 09:41] Alice: Budget update: The total funding is now set at $541,603.
[Day 1 10:17] Eve (Finance): Budget change: The authorized spending limit is now $1,684,675.
[Day 1 12:33] Bob: Rights cleared for the new song. We are using "Wild Midnight World".
[Day 1 14:33] Alice: We are locking the final budget at $1,073,223.
[Day 1 15:45] Frank (Legal): Team, regarding the venue: We are officially switching the location to Neo Valley Hall, Building C3.
[Day 1 16:59] Bob: Music choice change: The official theme song will be "Wild Midnight Fire".
[Day 1 18:49] Frank (Legal): Confirmed: The official date for the product launch is May 13, 2026.
[Day 2 09:10] Charlie: Let's finalize the location. We are going with Grand Valley Arena, Building A9.
[Day 2 10:52] Charlie: Finance has adjusted the budget cap. We now have $1,694,079.
[Day 2 13:56] Alice: We need to reschedule. The event will now happen on this date: March 4, 2026.
[Day 2 15:31] Charlie: Update on the place: Management wants us to use Blue Link Center, Building A4 instead.
[Day 2 17:39] Dave (PM): Logistics update: We will hold the event at Prime Harbor Arena, Building C4.
[Day 3 09:22] Bob: We need to reschedule. The event will now happen on this date: February 8, 2026.
[Day 3 12:35] Frank (Legal): The team voted on the song. The winner is "Eternal Summer Dreams".
[Day 3 15:34] Eve (Finance): The team voted on the song. The winner is "Urban Morning Soul".
[Day 3 17:55] Eve (Finance): We need to reschedule. The event will now happen on this date: May 24, 2026.

--- END OF SLACK HISTORY ---

Based on the chat history above, identify the FINAL agreed upon details for the launch event.
Please output the answer in the following format:
Place: <value>
Date: <value>
Budget: <value>
Song: <value>

Figure 26: Input example for the "Product Launch State Tracking" verification task. This sample illustrates the condition with an Update Count of 4 per key. Unlike the standard synthetic benchmark, updates here are embedded within a naturalistic, multi-turn team chat history (simulating a "Slack" export). The model must track the state changes of four specific entities (Budget, Song, Place, Date) amidst conversational noise and retrieve the final agreed-upon values at the end.

