# OpenReview forum: "Proactive Interference Reveals Working Memory Limits in LLMs Beyond Context Length"
_ICLR.cc/2026/Conference — Submitted to ICLR 2026_

### Official Review · Reviewer_Ue62 · 2025-10-27

**Soundness:** 2
**Presentation:** 2
**Contribution:** 2
**Rating:** 4
**Confidence:** 3

**Summary:**

LLMs are tested with PI-LLM, a setup that streams many co-referenced key–value updates and then asks for the final value. Across diverse models, accuracy falls roughly log-linearly as interference (more updates/keys/longer values) grows—even when total prompt length is held constant—implicating a working-memory-like interference limit rather than context size. Prompting or CoT doesn’t fix it; a simple “reset” hack helps only partially. The paper proposes an Interference Endurance Score (IES) and observes larger dense models resist interference better than smaller/MoE ones.

**Strengths:**

* Clean, controlled benchmark that isolates proactive interference (not search or sheer length) via co-referenced updates and a fixed-length control.
* Reveals a consistent, interpretable log-linear degradation pattern across many model scales, providing a quantitative handle on interference robustness.
* Demonstrates that current prompting and reasoning strategies are insufficient

**Weaknesses:**

The main concern is that the setting is extremely limited. The experimental design is clean, but it’s also very artificial. Because the task uses simple key–value updates, it’s hard to know whether the same interference appears in real settings.
* You could test a long article where an entity’s attributes change over time and see if the model retrieves the most recent one.
* A multi-turn chat or agent session, where a small profile field keeps being updated, would also make the setup feel more realistic.
* Showing that models with stronger interference in your test also make more “old value” mistakes in these realistic cases would make the results more convincing.

Most of the interventions the paper tests are just prompt variations, which doesn’t really address the underlying issue. If the problem is interference between parts of the context, there should be at least one experiment that changes how the model or system handles memory.(e.g., KV-cache resets/segmenting, retrieval-augmented state, local attention, active-experts in MoE).

The formatting of the figures could be improved (e.g. in figure 1, the 1*, 2*, 3*, 4* is hard to read).

**Questions:**

How is the fixed-length control enforced at the token level? The same number of key–value pairs doesn’t guarantee equal token length—please specify the tokenizer, any padding/truncation, and whether the final query length is held constant.

---

> ### Author Response · Authors · 2025-11-21
> **PI-LLM explains MRCR Difficulty and provide a Bridge to working memory definition of both human and llm.**
>
> Thank you for the thoughtful and constructive review. We appreciate the feedback on the ecological validity of PI-LLM and the suggestions for exploring realistic settings (e.g., multi-turn chat/entity updates). We address these crucial points and the call for system-level interventions below, with new experiments and analysis.
>
> ---
> **1.We added a verification experiment for PI in naturalistic setting.**
> Instead of synthetic lists, we generated meeting transcripts where 4 specific project details (Place, Date, Budget, Song) are updated repeatedly within the conversation flow. We tested gpt-4.1-nano (logscale update counts: 4 – 256).
>
> **Acc (update count)**
> - 0.8 (n=4)
> - 0.7 (n=8)
> - 0.4 (n=16)
> - 0.6 (n=32)
> - 0.4 (n=64)
> - 0.2 (n=128)
> - 0.2 (n=256)
>
> **Details in Figure 25&26 of Appendix** :
> Consistent Decline: The log-linear decline in accuracy persists even in this natural setting (dropping from 80% to 20%).
> Error Analysis: The model primarily retrieves outdated previous updates rather than hallucination.
> This confirms that Proactive Interference is a fundamental limitation that holds true in natural discourse beyond synthetic tasks.
>
> **2. Connection to MRCR Test**
> PI-LLM is a **controlled, cognitively valid, and minimal task** that models the core difficulty of **Multi-Round Co-Reference Resolution (MRCR)**, a standard, long-context challenge (e.g., Gemini 3 Pro, GPT-5). **By stripping away retrieval/haystack factors, PI-LLM isolates the core binding/unbinding bottleneck that underlies MRCR.** MRCR simulates multi-turn interactions where an entity is repeatedly updated, causing SOTA model failures. PI-LLM provides a clean, quantitative probe for **this core mechanism**, **opening up paths for mechanistic understanding to solve MRCR more efficiently.**
>
> *MRCR benchmarks in major model releases*
> - Google (2025). *A new era of intelligence with Gemini 3.* Google Product Blog.
> - OpenAI (2025). *Introducing GPT-5 for developers.* OpenAI.
>
> ---
> 3. **Clarification on fixed‑length control and tokenizer**
> In Expm.2 (fixed length, varying interference), we control input length by changing only the final query text while keeping the updating stream untouched. As the majority of the inputs are the updating sequence, the influence to overall input length is negeligable (<0.1%).
>
> **Token-level verification** For example, while 46 keys are updated, we could remove 20 keys from the query list in the instruction part. As a result, the model would only track 26 keys instead of 46 keys. The original full prompt is 47,077 tokens. Depending on the tokenizer, this corresponds to a change of 20–40 tokens. Using the high bound (40 tokens), the relative change is 40 / 47 077 ≈ 0.085%; Thus, while the tracking load (number of keys to retrieve) varies substantially, the input length remains effectively constant and is essentially agnostic to the specific tokenizer used.
>
> ---
> 4. **Mechanistic analysis beyond “just prompts”**
> (Chapter 7, Figure 7). Motivated by the reviewer’s request for more than prompt variations and building on prior work on entity-binding representations in LLMs, we added a **mechanistic probe of PI-LLM itself**
>
> We construct a context where a single key is sequentially updated to 25 distinct values and query “What is the *n*-th value of key1?” for \( n = 1,2,3...... 25 \). For each query, we extract the residual-stream activation at the last input token at every layer of Llama-3.2-1B and perform PCA per layer over the 25 query-position representations.
>
> We find that:
> - At mid-layers, the PCA plots show 25 points arranged along a smooth trajectory, with early vs. late values separated.
> - As n increases and more bindings accumulate, neighbouring values move closer and begin to overlap, indicating **progressive compression** of the “binding subspace”: internal representations no longer cleanly separate different values for the same key.
> - Crucially, the crowded region in PCA coincides with the behavioral failure regime: the late-( n ) points collapse into an overlapping cluster exactly where retrieval errors concentrate. The **geometric compression and the error pattern aligned.**
>
> This experiment connects the PI-LLM performance curve to the **entity-binding geometry** described in interpretability work (Dai et al., 2024; Feng et al., 2023), and suggests the direction for mitigation: architectures or training schemes that preserve better separation of multi-binding subspaces should improve interference resistance. Our goal is to make the failure mode **mechanistically interpretable**. (Appendix Fig.24)
>
> *Entity-binding interpretability*
> – Dai, Q., Geva, M., Geiger, A., & Goldberg, Y. (2024). *Representational Analysis of Binding in Language Models.* EMNLP 2024.
> – Feng, J., et al. (2023). *How Do Language Models Bind Entities in Context?* arXiv:2310.17191.
>
> We are happy to improve figure readability in the revised manuscript by replacing the $1^*, 2^*$ with superscripts next to the updated values.

---

### Official Review · Reviewer_eEkx · 2025-10-28

**Soundness:** 2
**Presentation:** 2
**Contribution:** 2
**Rating:** 4
**Confidence:** 3

**Summary:**

The paper investigates into a particular factor that impacts LLM performance: limited anti-interference capacity, where the paper shows that LLM performance degrades in a log-linear way as interferences increase.

In the experiments, the authors show that this increase is disentangled with length increase (Section 3), thus makes it an independent causal factor. The tests have been carried over numerous LLMs and they show similar log-linear decrease trend. Besides, the authors have investigated into ways to mitigate the issue including various prompt changes as well using CoT. The improvement was only minimum and the authors show that the problem persists well.

**Strengths:**

Understanding the LLM limitation is surely an important topic, particularly if novel insights are brought into the community. The paper identified the anti-inference capacity as an LLM performance limitation; most importantly, the paper has shown that the inference factor is independent of context length factor to impact LLM performance.

The paper has demonstrated the results over various LLMs to show that it is a general problem for current LLMs; besides, the paper has investigated into different prompt strategy including CoT to mitigate the issue and show that the problem persists.

**Weaknesses:**

I don't doubt the novelty in this work, nevertheless, I would encourage the authors to include a Related Work to show the connections that the paper has with existing research.

The results demonstrated in the paper, well kind of novel, is not surprising: the LLM performance is not conditioned on the length of the context but also the problem difficulty; for the later case, the community has identified quite early on that the LLM memory can be one of such problem difficulty (in this sense, the current paper is well related to these lines of works). While prompt strategy has been investigated, we also notice that more involved architectures (LLM workflow, or agents) are not investigated in the paper; sure that it is not directly linked with the interference factor that limits the LLM performance, but it is important to show that these problems can/cannot be solved alternatively, for the current paper, it reads like it is not solvable which I believe is not true.

**Questions:**

None

---

> ### Author Response · Authors · 2025-11-21
> **From Human Working Memory to LLMs: Proactive Interference Explains MRCR Long-Context Failures**
>
> We thank the reviewer, the comments helped pushing our positioning relative to prior work, discover the link between MRCR and LLM entity binding from LLM interp line of study.
>
> ---
> We agree that the high‑level intuition that greater memory load or task difficulty harms accuracy is well established. Our contribution is to turn this intuition into a **cognitive-science verified, quantitative comparison**. The "surprise" is that LLM behave as if they lack an effective ‘ignoring’ mechanism which is present in human.
>
> Previous study on cognitive test on LLM (AAAI24) https://arxiv.org/abs/2305.03731conduct toy level synthetic task. **In contrast, **our test compares the working-memory-like binding capacity of humans and LLMs using a realistic long-context retrieval task.**
>
> PI‑LLM isolates a configuration where the same key/entity is repeatedly rebound to different values (A→B₁, A→B₂, …, A→Bₙ) and retrieve a specific binding (e.g.last value). This test **mirroring classic proactive interference paradigms** in human working‑memory research to study how human **keep track of binding informations.**
>
>
> **Reference**
>
> Working Memory Capacity of ChatGPT: An Empirical Study (AAAI 2024)
>
> ---
> **2. New PCA experiment, mechanistic interpretation, and path to mitigation**
>
> Motivated by the reviewer’s comment to consider potential mitigation directions, and building on prior work on entity-binding of LLM's representation (where bindings such as “A associated with B” occupy a structured, decodable subspace), we add a mechanistic probe connecting binding geometry to PI-LLM retrieving performance.
>
> **(Chapter 7)** We set PI-LLM test with key=1 and update =25, and query “What is the *n*-th value of key1?” for \( n = 1,2,..... 25 \). For each query, we extract the **residual-stream activation** at the last input token at every layer of Llama-3.2-1B and perform PCA per layer over the 25 query-position representations.
>
> We show that:
> - At mid-layers, the PCA plots show 25 points arranged along a smooth trajectory, with early vs. late values separated.
> - For larger n,  more bindings accumulate and neighbouring values move closer and begin to overlap, indicating **progressive compression** of the “binding subspace”: internal representations no longer cleanly separate different values for the same key.
> - **Crucially**, The layer-wise **PCA geometry and the behavioral errors are tightly aligned**: the region where different bindings collapse in the residual space is exactly where retrieval accuracy drops.
>
> This experiment bridges the PI-LLM performance curve to the **entity-binding geometry** described in interpretability work (Dai et al., 2024; Feng et al., 2023), and suggests the direction for mitigation: architectures or training schemes that preserve better separation of multi-binding subspaces should improve interference resistance. (Appendix Fig.24)
>
> In the revision, we add Related Work that situates PI‑LLM within:
> (i) entity‑binding studies in LLM interpretability and Human Cognitive Study.
> (ii) long‑context benchmarks: MRCR.
>
> *Entity-binding interpretability*
> – Dai, Q., Geva, M., Geiger, A., & Goldberg, Y. (2024). *Representational Analysis of Binding in Language Models.* EMNLP 2024.
> – Feng, J., et al. (2023). *How Do Language Models Bind Entities in Context?* arXiv:2310.17191.
>
> ---
> **3. Binding explains MRCR**
>
> PI‑LLM provide a minimal, controlled abstraction of the core difficulty in MRCR long‑context benchmarks. MRCR tasks, now standard in major model reports (e.g., Gemini 3 Pro, GPT‑5), simulate realistic multi‑turn interactions where a single entity is referenced and updated multiple times, and the model must reproduce a specific binding answer. Empirically, MRCR shows that models often fail once the same entity has been re‑bound many times, even though all necessary information is in context.
>
>
> PI‑LLM removes haystack/search factors and keeps only the multi‑coreference binding–unbinding core: maintaining multiple rebindings of one entity and then retrieving a specific one. By varying only the number of co‑referenced updates and measuring the resulting log‑linear accuracy drop, PI-LLM provides a clean capacity probe of the same interference-driven difficulty that MRCR exposes.
>
> We emphasise in the revised narrative that PI-LLM uses a standard cognitive-science interference manipulation, links it to **entity-binding** representations of LLMs, and offers a mechanistic explanation for MRCR challenges.
>
> *Reference*
> **MRCR benchmarks in major model releases**
> - Google (2025). *A new era of intelligence with Gemini 3.* Google Product Blog.
> - OpenAI (2025). *Introducing GPT-5 for developers.* OpenAI Blog.
>
> ---
> Pi-LLM (i)turns a strict cognitive science experiment that bridge LLM interp to mechanistically interpret the MRCR test from an entity binding perspective. And( ii) provide a clean capacity probe that compare to human working memory   with LLMs. (iii)extend prior work on LLM–human comparative cognition study,

---

> > ### Comment · Reviewer_eEkx · 2025-11-22
> >
> > I thank the authors for their comments. Particularly point out the most related works in the first point as well as the relevancy in the third point.
> >
> > The second point brings values on how the problems occur which is valuable. Nevertheless, I continue to believe that largely solve the problem via advanced prompting or using agentic framework can bring further value to the paper: it shows what LLMs miss and how that can be still solved with current LLM landscape.

---

> ### Author Response · Authors · 2025-11-23
> **Proactive Interference as a Mechanistic Bridge for LLMs and Humans**
>
> Thank you for the timely comment. Your suggestion led us to examine concurrent work evaluating agentic frameworks on PI-LLM and MRCR
>
> To address your concern, we examined **(i) concurrent work on agentic systems**  (ii) our **own experiments with very aggressive chain-of-thought / Reasoning settings and prompt variants**, and (iii) **external evaluation** and comment on MRCR from OpenAI and DeepMind team. Together, these results indicate that the **phenomenon is robust** and not easily resolved by current prompt or agentic methods.
>
> ---
>
> ### 1. Agentic frameworks tested on PI-LLM and have only marginal effect
>
> Concurrent work by Li et al. (2025) directly evaluates agentic frameworks on **PI-LLM and MRCR** and show that **agents also struggle on PI-LLM and MRCR**. They then introduce Sculptor, an improved agent framework with Active Context Management (ACM) tools that let an LLM rewrite and curate its own conversation history.
>
> They evaluate models such as Claude-4-Sonnet, GPT-4.1, and DeepSeek-V3 on PI-LLM and related long-context benchmarks. On PI-LLM, Sculptor yields only modest average gains of ≈6–7 percentage points for Claude-4-Sonnet and GPT-4.1, while DeepSeek-V3 shows **essentially no improvement** (−0.1 percentage points).
>
> Table : Sculptor (Li et al., 2025) PI-LLM accuracy by update count and context length.
>
> | Model           | Method       |  4/1K |  8/2K | 16/4K | 32/8K | 64/16K | 128/32K | 256/64K |   Avg |
> | --------------- | ------------ | ----: | ----: | ----: | ----: | -----: | ------: | ------: | ----: |
> | Claude-4-Sonnet | Baseline     | 99.13 | 95.65 | 92.17 | 84.78 |  81.74 |   65.22 |   69.57 | 84.04 |
> |                 | w/ Agent Tools | 90.43 | 91.74 | 98.26 | 92.17 |  91.74 |   87.39 |   77.83 | 89.94 |
> | GPT-4.1         | Baseline     | 96.96 | 91.30 | 79.57 | 67.83 |  63.04 |   63.91 |   50.43 | 73.29 |
> |                 | w/ Agent Tools | 92.17 | 89.13 | 93.04 | 83.91 |  76.09 |   64.35 |   60.43 | 79.87 |
> | DeepSeek-V3     | Baseline     | 95.22 | 85.65 | 70.00 | 63.91 |  33.04 |   32.17 |   21.74 | 57.39 |
> |                 | w/ Agent Tools | 73.91 | 90.00 | 79.13 | 37.39 |  53.04 |   55.65 |   11.74 | 57.27 |
>
> Overall, even with sophisticated active context management and tool-augmented agents, Sculptor improves PI-LLM accuracy modestly and the **decline patter of performance persist.**
>
> [https://arxiv.org/abs/2508.04664v2](https://arxiv.org/abs/2508.04664v2)
>
> **Reference**
>
> Li et al. (2025). *Sculptor: Empowering LLMs with Cognitive Agency via Active Context Management.*
>
> ---
>
> ### 2. Prompt engineering and **“unlimited”** reasoning also have limited effect
>
> Motivated by the reviewer’s comment, we also add test to see whether very aggressive chain-of-thought / Reasoning settings with **unlimited reasoning budget** can solve binding.
>
> We added an evaluation of GPT-5 on a middle-interference PI-LLM configuration (Experiment 1, 197 updates, 46 query keys). We compare a “medium effort” and a “high effort” setting, which differ in Reasoning configuration (number of Reasoning tokens and deliberation time):
>
> **GPT-5 Proactive-Interference (PI) test on PI-LLM (197 updates, 46 keys).**
>
> | Submission    | CoT tokens | Deliberation time | Correct / Total | Accuracy |
> | ------------- | ---------: | ----------------- | --------------: | -------: |
> | Medium effort |      15.9k | 6 min 25 s        |         18 / 46 |   39.1 % |
> | High effort   |      39.9k | 10 min 00 s       |         33 / 46 |   71.7 % |
>
> Even in the high-effort Reasoning setting, where GPT-5 produces ≈**39.9k CoT** tokens and answers latency of ~10 minutes, accuracy reaches only 71.7% (33/46), up from 39.1% (18/46) at medium effort with 15.9k reasoning tokens generated. This shows that SOTA models and very aggressive Reasoning modes **partially mitigate interference**, but they **do not eliminate** it, and performance **does not trend cleanly toward ceiling** even for GPT-5 with high-Reasoning, **The decline pattern persist.**
>
> In addition, those result might help  we evaluated variation of prompt engineering:
> **Prompt engineering and meta-prompts.**
>
> * A “Relevance meta-prompt” **(Fig. 5 in the paper)** where the **LLM is first asked to analyze the task** and **design own analysis** for PI-LLM **before conducting the task.**
>
> **Across these prompt variants, the decline curve is essentially unchanged** and closely aligns with the baseline performance. Different prompts shift the curve slightly but do not change its approximately log-linear shape or the fact that errors are dominated by retrieving outdated bindings rather than hallucinating new values.
>
> Taken together, these results indicate limitations of prompt variation and extensive Reasoning in solving this problem: they **can improve performance by several percentage points, but they do not change the underlying mechanism of proactive interference in binding process.**
>
>
> **Reference**
>
> (Fig. 5 in the paper)
>
> ---
> **Part1/2 End.

---

> ### Author Response · Authors · 2025-11-23
> **Proactive Interference as a Mechanistic Bridge for LLMs and Humans/Part II**
>
> ###Part2-continue
> ---
> ### **3. External evidence from OpenAI and DeepMind: MRCR results**
>
>
> The hardness of this regime is independently acknowledged in public long-context benchmarks. OpenAI’s MRCR benchmark and Google DeepMind’s MRCR v2 results provide converging evidence that long-context retrieval with multiple coreference (multiple binding) remains challenging even for advanced reasoning models.
>
> They focus on multi-reference/co-reference retrieval under heavy distraction, which is precisely **the regime that PI-LLM ablates and isolates.**
>
> **OpenAI MRCR.**
> In the GPT-4.1 release note, OpenAI reports performance on their MRCR benchmark and explicitly writes:
>
> **“But the task remains hard—even for advanced reasoning models.** We’re sharing the eval dataset to encourage further work on real-world long-context retrieval.” ---------OpenAI Releasing Note
>
> [https://openai.com/index/gpt-4-1/](https://openai.com/index/gpt-4-1/)
>
> **Gemini 3 Pro model card (MRCR v2, 8-needle).Performance**
>
>
> Table: MRCR v2 (8-needle) long-context performance from the Gemini 3 Pro model card.
>
> |                                     | Gemini 3 Pro | Gemini 2.5 Pro | Claude Sonnet 4.5 |       GPT-5.1 |
> | ----------------------------------- | -----------: | -------------: | ----------------: | ------------: |
> | MRCR v2 (8-needle) – 128k (average) |        77.0% |          58.0% |             47.1% |         61.6% |
> | MRCR v2 (8-needle) – 1M (pointwise) |        26.3% |          16.4% |     not supported | not supported |
>
> [https://deepmind.google/models/gemini/pro/](https://deepmind.google/models/gemini/pro/)
>
> Thus, even cutting-edge long-context models (Gemini 3 Pro, GPT-5.1, Claude Sonnet 4.5) display decline of accuracy in co-reference.
>
>
> **a).What MRCR probes**. MRCR targets the **same regime as PI-LLM**: multi-reference / co-reference retrieval under heavy distraction, i.e., **repeated rebinding of entities**.
>
> **b).In MRCR, the 8-needle setting corresponds closely to PI-LLM Experiment 1** with 8 updates per key (i.e., 8 co-referenced “needles” to disentangle). PI-LLM then ablates and extends this regime by systematically increasing the number of co-referenced updates up to 400.
>
>
> ---
> ### **4. Mitigations Experiment (both prompt variants & agentic pipelines)**
> We agree that attempts at mitigation may provide useful perspective on what is missing in current LLMs.
>
> In the paper we conducted a experiment using targeted "Mock QA reset" prompt **(see Fig. 15& 16;  "Mock QA reset" orange line)(and Appendix (Section. F.3): “Mock-QA-Reset”)**, which **boosts accuracy** at several interference levels.
>
> The Mock-QA-Reset prompt frames earlier text as part of already completed sub-task, inducing the model to treat them as completely out-of-scope for the current query.  This partially mitigates proactive interference and yields larger accuracy gains than reasoning models with extensive token budgets.
>
> Concurrent agent work (Li et al., 2025; Sculptor) improves performance on PI-LLM and MRCR. Mechanistically, these gains come from active context editing: the agent splits the dialogue and **folds/temporarily delete earlier fragments** (replacing them with short placeholders) so the interfering updates no longer participate in subsequent reasoning. When folding is too aggressive, accuracy drops at low update counts **(useful information is removed)**.
>
> Thus, their **agents improve PI-LLM chiefly by rewriting the context to remove interference**, not by making the base model robust to interference within a fixed context.
>
> Importantly, in both attempt, the qualitative **decline pattern remains unchanged**
>
> Reference
>
> Li et al. (2025). Sculptor: Empowering LLMs with Cognitive Agency via Active Context Management.
>
> ---
>
> ## Summary.
>
>
> Below we summarize evidence that:
>
> 1. **Agentic frameworks already tested on PI-LLM / MRCR yield only marginal gains**,
> 2. **Prompt engineering and “unlimited” Reasoning do not remove the proactive-interference pattern**, even for state-of-the-art models such as GPT-5, and
> 3. **Independent MRCR results from major model providers** confirm that this regime remains hard even for advanced reasoning models.
>
> These observations, taken together, support the decision to focus this paper on mechanistic analysis and a cognitive-science-based study of LLMs. This Paper is to make the failure mode mechanistically interpretable:
>
> **We believe “to solve it, we first need to know more about it.”**
>
> This paper aims to **investigate LLM working memory in a way that is grounded in cognitive science**, following prior work on LLM–human comparisons (e.g., *Working Memory Capacity of ChatGPT*, AAAI 2024). Our goal is to:
>
> * Isolate the root cause of multi-update / co-reference challenges, and
> * Study the nature of LLM “working memory” in direct **parallel with human performance paradigms.**

---

### Official Review · Reviewer_zn3J · 2025-10-31

**Soundness:** 3
**Presentation:** 3
**Contribution:** 2
**Rating:** 6
**Confidence:** 3

**Summary:**

This paper investigates how information retrieval in LLMs is affected by intra-context interference, adapting the proactive interference (PI) paradigm from cognitive science. The authors propose PI-LLM, an evaluation where models must recall the latest value among sequentially updated, co-referenced key–value pairs. Results show a log-linear decline in retrieval accuracy as interference accumulates, with models often recalling outdated values. Prompt-based mitigation fails to resolve the issue, indicating a working memory–like limitation in LLMs. By removing irrelevant context (“haystack”) and directly measuring interference, PI-LLM provides a principled framework to assess and improve LLMs’ information disentanglement and memory control.

**Strengths:**

1. Clear and well-written: The paper is logically structured and easy to follow, with clear motivation and experimental design.

2. Valuable and insightful finding: The discovery of log-linear interference effects offers deep insights into LLM working memory limits and provides meaningful guidance for future model development.

**Weaknesses:**

1. No discussion with a very related work (motivation and discovery): https://arxiv.org/abs/2502.05252.  This paper also discusses how to insert noise in the context and find something very similar to log-linear degradation.

2. Do not provide executable insights on how to improve LLM training in the discussed problem.

**Questions:**

In the weakness.

1. Are models making advancements in the discussed problem? For example, Llama 2, Llama 3, Llama 4/Qwen3/Deepseek —how is the trend?

2. Does model architecture make a difference? For example, xLSTM and Mamba are compared to transformers.

---

> ### Author Response · Authors · 2025-11-21
> **Stable pattern from 1B Models to GPT-5: A Mechanistic Bridge Between Bechmark and LLM Interpretability**
>
> We thank the reviewer for the thoughtful and positive assessment.
>
> ### 1. Are stronger models and more inference effort solving the problem? (GPT-5)
>
> To address the reviewer's question about trends and whether stronger models or heavier Reasoning resolve the issue, we added evaluation of GPT-5 on a middle-interference PI-LLM configuration (Experiment 1,197 updates, 46 query keys).
>
> **GPT-5 Proactive-Interference (PI) Test**
>
> | Submission     | CoT tokens | Deliberation time | Correct / Total | Accuracy |
> |----------------|-----------:|-------------------|----------------:|---------:|
> | Medium effort  |    15.9k   |  6 min 25 s       |        18 / 46  |  39.1 %  |
> | High effort    |    39.9k   | 10 min 00 s       |        33 / 46  |  71.7 %  |
>
> Even in the high-effort Reasoning setting, where GPT-5 produces ≈39.9k CoT tokens and latency up to 10 minutes, accuracy reaches only **71.7% (33/46)**, up from **39.1% (18/46)** at medium effort.
>
> This shows that SOTA models and more aggressive Reasoning Settings **partially** mitigate interference, but they do **not** eliminate it, and performance does not trend cleanly toward ceiling for GPT-5. This leads to next mechanistic explanation section.
>
> ---
> ### 2. Related work
>
> GSM-Infinite.We appreciate the pointer to GSM-Infinite (Zhou et al., 2025) and agree that it is related at the level of motivation: both works study how performance degrades as more “stuff” is packed into the context. We have added GSM-Infinite to the beginning of Chapter 7.
>
> ---
>
> ### 3. (Chapter7) New PCA experiment, mechanistic interpretation, and implications for mitigation
>
> Motivated by reviewer’s request for more “executable” insight and building on prior work on entity-binding representations in LLMs, we perform a **mechanistic probe of PI-LLM itself**. (Chapter7 Fig.~7)
>
> We construct a context where a single key is sequentially updated to 25 distinct values and issue queries “What is the n‑th value of key1?” for n = 1,…,25. For each query, we extract the residual‑stream activation at the last input token at every layer of Llama‑3.2‑1B and run PCA per layer.
>
> We find (Appendix Fig.~24):
>
> - At mid-layers, the PCA plots show all points arranged along a smooth trajectory, with early vs. late values separated.
> - As \( n \) increases and more bindings accumulate, neighbouring values move closer and begin to overlap, indicating **progressive compression** of the “binding subspace”: internal representations no longer cleanly separate different values for the same key.
> - Crucially, the crowded region in PCA coincides with the behavioural failure regime: the late-( n ) points collapse into an overlapping cluster exactly where retrieval errors concentrate. **The geometric compression and the error pattern are tightly aligned.**
>
> This experiment bridges the PI-LLM performance curve with the **entity-binding geometry** studied in interpretability work (e.g., Dai et al., 2024; Feng et al., 2023), and suggests a concrete direction for future mitigation: architectures or training schemes that preserve better separation of multi-binding subspaces should improve interference resistance.
>
> Our intention is to make the failure mode **mechanistically interpretable**: “to solve it, we first need to fully understand it.”
>
> *Entity-binding interpretability*
> – Dai, Q., Geva, M., Geiger, A., & Goldberg, Y. (2024). *Representational Analysis of Binding in Language Models.* EMNLP 2024.
> – Feng, J., et al. (2023). *How Do Language Models Bind Entities in Context?* arXiv:2310.17191.
>
> ---
>
> ### 4. How PI-LLM explains MRCR long-context failures and clarifies scope
>
> PI-LLM is designed as a **minimal, controlled abstraction** of the core difficulty behind MRCR-style long-context benchmarks.
>
> MRCR tasks, now standard in major model reports (e.g., Gemini 3 Pro, GPT-5), simulate realistic multi-turn interactions where a single entity is referenced and updated multiple times, and the model must retrieve a specific binding. PI-LLM isolates exactly this **multi-coreference, binding/unbinding** problem:
>
> - We remove haystack/search factors and keep only the **co-referential binding core**: maintaining multiple rebindings of one or more entities and then retrieving a particular one.
> - By varying only the **number and organisation of co-referenced updates** and measuring the resulting approximately log-linear accuracy drop, PI-LLM provides a **quantitative capacity probe** for the same underlying difficulty that MRCR exposes.
>
>
> Using a strict, cognitive-science verified PI test, we show that LLMs **share several key constraints with human working memory** and we use internal binding representations from the LLM interp studies to reveal the mechanics of this task.
>
>
> *References*
> **MRCR benchmarks in major model releases**
> - Google (2025). *A new era of intelligence with Gemini 3.* Google Product Blog.
> - OpenAI (2025). *Introducing GPT-5 for developers.* OpenAI Blog.

---

### Official Review · Reviewer_LZv7 · 2025-10-31

**Soundness:** 2
**Presentation:** 1
**Contribution:** 2
**Rating:** 4
**Confidence:** 4

**Summary:**

LLMs are known to struggle when conflicting evidence is present in the context. This has been extensively studied in the past in relation to RAG (retrieval augmented systems). This paper proposes a task to systematically evaluate the same. The context consists of key value pairs, with same key potentially having differing values in the same context. The LLM is supposed to find the value associated with the last occurrence of a key. The work shows that increasing such conflicts leads to a consistent drop in the accuracy.

**Strengths:**

- THe experiments are thorough. The authors clearly specify the prompts they used to ensure that the model follows the task as desired.
- The control for various confounders like length.
- The capacity analysis is nice.

**Weaknesses:**

- It is a well-documented fact in the RAG literature that intra-context conflicts cause performance degradation (see https://arxiv.org/abs/2507.21544, https://arxiv.org/abs/2504.13079v2). In fact, prior works have shown that when conflicting evidence is present in context, models tend to rely on parametric knowledge biases rather than the retrieved evidence (https://arxiv.org/abs/2402.07867, https://aclanthology.org/2022.emnlp-main.146.pdf).

- Given these studies, it is unclear what the proposed benchmark adds—whether it identifies a genuinely new failure mode or deepens understanding of existing, well-known interference phenomena.

- Relation to real-world scenarios: In realistic settings, conflicting evidence typically co-occurs with confounders such as the confidence of retrieved snippets or the credibility of their sources. These factors often dominate which information an LLM uses in its final response. The proposed benchmark abstracts away such factors, making it difficult to extract actionable insights for practitioners. It largely remains a toy key–value benchmark with limited ecological validity.

- Suggestions: Future versions of the work could incorporate more nuanced and realistic setups—e.g., multi-hop reasoning or information extraction under conflicting evidence—to better connect with real-world retrieval challenges.

- Minor: The plots can be made much more cleaner in the main paper, with only key curves being shown and rest deferred to the appendix. In general, the manuscript needs significant more efforts in better presentation.

**Questions:**

See the weakness section

---

> ### Author Response · Authors · 2025-11-21
> **From Human Proactive Interference to LLMs: A Cognitive Binding Probe Linking Mechanistic Analysis and MRCR‑Style Tests**
>
> Thank you for the insightful review. For the thoughtful and detailed comment. We address concerns about novelty, relation to real‑world scenarios, and actionability below, and have updated the manuscript accordingly.
>
> ---
>
> ### 1. Main contribution and novelty compare to RAG conflict work
>
> Prior work on RAG with conflicting evidence shows that intra‑context conflicts degrade performance and can cause models to rely on parametric knowledge instead of retrieved evidence;
>
> And a dedicated Related work paragraph discussing these studies is updated to the manuscript.
>
> We agree that real‑world setups involve many additional factors. Our goal in PI‑LLM is to provide a minimal, controlled abstraction of the core multi‑binding difficulty and provide a mechanistic explanation of MRCR‑style failures as a working‑memory–like binding/unbinding bottleneck, and to link this to entity‑binding in LLM Interp line of studies. PI‑LLM isolates a configuration where the same key/entity is repeatedly rebound to different values (A→B₁, A→B₂, …, A→Bₙ) and the model must retrieve a specific binding (e.g., the n‑th or last value). Under control of context length and other confounders, we observe a log‑linear decline in accuracy and characteristic old‑value errors, directly mirroring classic proactive interference paradigms in human working‑memory research. Thus, PI‑LLM acts as a working‑memory–like binding capacity test, a classical cognitive test for Human WM studies.
>
>
> **RAG conflict evidence**
> - Lee, J., Jang, J., & Kim, G. (2025). *MAGIC: A Multi-Hop and Graph-Based Benchmark for Inter-Context Conflicts in Retrieval-Augmented Generation
> - Wang, H., Prasad, A., Stengel-Eskin, E., & Bansal, M. (2025). *Retrieval-Augmented Generation with Conflicting Evidence.*
> - Zou, W., et al. (2024). *PoisonedRAG: Knowledge Corruption Attacks to Retrieval-Augmented Generation.* arXiv:2402.07867.
> - Chen, H.-T., Zhang, M. J. Q., & Choi, E. (2022).
> *Rich Knowledge Sources Bring Complex Knowledge Conflicts: Recalibrating Models to Reflect Conflicting Evidence.* EMNLP 2022.
>
> ---
>
> ### 2. (NEW Chapter7) Mechanistic interpretation and new PCA experiment
>
>
>
> Inspired by the review we building on prior work on entity‑binding representations in LLM interp, which shows that bindings such as “A associated with B” occupy a structured, decodable subspace, we add a mechanistic probe of PI‑LLM itself.
>
>
> We construct a context where a single key is sequentially updated to 25 distinct values and issue queries “What is the n-th value of key1?” for $n = 1,\dots,25$. For each query, we extract the residual-stream activation at the last input token at every layer of Llama-3.2-1B and run PCA per layer. (Chapter7 Fig.~7)
>
> At mid‑layers, the PCA plots (Appendix Fig.~24) show another 16 query test, points arranged along a smooth trajectory, with early vs. late values separated. As $n$ increases, clusters corresponding to neighbouring values move closer and begin to overlap, indicating that the binding subspace is progressively compressed: internal representations no longer cleanly separate different values for the same key, matching the observed accuracy drop and $n\!\pm\!1$ confusions. A full‑layer scan shows that this compression pattern is robust across depth.
>
> **Entity-binding interpretability**
> - Dai, Q., Geva, M., Geiger, A., & Goldberg, Y. (2024). *Representational Analysis of Binding in Language Models.* EMNLP 2024.
> - Feng, J., et al. (2023). *How Do Language Models Bind Entities in Context?* arXiv:2310.17191.
>
> ---
>
> ---
>
> ### 3.  How PI-LLM explains MRCR challenges
> PI‑LLM is a minimal, controlled abstraction of the core difficulty in MRCR‑style long‑context benchmarks. MRCR tasks, now standard in major model reports(Gemini 3 PRO,GPT5), simulate realistic multi‑turn interactions where a single entity is referenced and updated multiple times, and the model must reproduce a specific earlier answer or the most recent state. Empirically, MRCR shows that models often fail once the same entity has been re‑bound many times.
>
> Consistent with MRCR results, PI-LLM remains challenging for SOTA models: GPT5 high-effort reasoning settings (≈39.9k Reasoning tokens and ≈10 minutes latency), reaches 71.7% accuracy (33/46), and  39.1% (18/46) at medium effort (Appendix Table 3).
>
>
> PI‑LLM removes haystack/search factors and keeps only the multi‑coreference binding–unbinding core: maintaining multiple rebindings of one entity settings. By varying only the number of co‑referenced updates PI‑LLM provides a quantitative measure of the same underlying difficulty that MRCR exposes. We emphasize in the revised manuscript that PI‑LLM is meant to explain MRCR failures mechanistically and to provide a clean capacity probe. and the performance of GPT5 in PI-LLM is added in the appendix.
>
>
> ---
>
> **MRCR benchmarks in major model releases**
> - Google (2025). *A new era of intelligence with Gemini 3.* Google Product Blog.
> - OpenAI (2025). *Introducing GPT-5 for developers.* OpenAI Blog.

---

### Author Response · Authors · 2025-11-29
**From Human Proactive Interference to LLMs: A Cognitive Lens on Long-Context Binding Limits**

Dear Area Chair,

We thank the reviewers for their feedback, which inspired us to sharpen the paper's positioning.

This rebuttal adds **substantial new experimental evidence** that was not included in the original submission — including a mechanistic PCA analysis (Section 7).

---

Our rebuttal clarifies that this work belongs to **Cognitive Science & Mechanistic Interpretability**, not RAG benchmarking. We investigate a **"Multiple Binding Bottleneck"**—a continuation of research into LLM working memory and entity binding mechanisms.

Our goal is to study the nature of LLM “working memory” in direct parallel with human performance paradigms, following prior work on LLM–human comparisons (e.g., *Working Memory Capacity of ChatGPT: An Empirical Study*, AAAI 2024).

---

**1. The "Ignoring" Mechanism (Human vs. LLM)**

Building on cognitive paradigms, our results quantify a key divergence: human working memory can **actively inhibit/discard** outdated bindings to reduce interference, whereas LLMs **accumulate** them. This follows a specific **log-linear decay law** that standard benchmarks miss. We benchmark against **three verified human cognitive interference phenomena** to provide a rigorous comparative lens.

---

**2. Distinction from RAG & Link to Interpretability**

RAG conflicts center on *selecting* evidence. In contrast, PI-LLM isolates *binding* (A→B₁…A→Bₙ) in a **fixed-length setting where the model’s answer token attends to all A→B updates within the same context window (no retrieval or truncation)**, explicitly separating binding from retrieval.

The **new PCA analysis(Section 7)** indicates that the failure is driven by the **progressive compression of the entity-binding subspace** as updates accumulate. This suggests the issue has a **representational (internal capacity) root**, rather than something that can be solved by prompt design alone, directly connecting to ongoing work in **LLM mechanistic interpretability**.

---

**3. Ecological Validity, Robustness, and Mitigation**

* **Realism:** We link PI-LLM to **MRCR (multi-round co-reference resolution)** failures reported in Gemini 3-PRO / GPT-5 technical reports and observe the same log-linear interference pattern in naturalistic meeting transcripts.
* **Hardness / Failure of Mitigation:** The bottleneck is robust. Even **GPT-5**, configured with **High Effort Reasoning** (substantially increased inference-time compute) and generating ~40k CoT tokens, achieves **71.7%** accuracy—compared to **39.1%** under *Medium Effort*. Consistent with this, our experiments with prompt variations and concurrent agentic studies show only marginal gains (+6–7%): the qualitative decline pattern remains unchanged. This provides evidence for an **architectural limit** that scale and prompting alone cannot fully resolve.

---

**Summary**

This work offers a **mechanistic, cognitively validated** explanation for why LLMs struggle on co-reference tasks.

We find that the same log-linear interference pattern remains **stable across model scales (from ~0.6B to ~600B+ SOTA models)**,up to GPT-5, and data shows it persists even in agentic settings. Taken together with the **mechanistic evidence** (PCA) supports the view that it reflects an underlying representational bottleneck rather than a problem that can be easily removed by prompt engineering.

The new evidence (PCA, MRCR link) bridges **LLM interpretability**, **real-world performance**, and **cognitively grounded measurements** of working memory.

---

### Meta-Review · Area_Chair_mqeL · 2026-01-13

**Summary:**

Reviewers broadly agreed that the paper identifies a real and systematic failure mode of LLMs—log-linear degradation under intra-context interference—that is robust across models, scales, and prompting strategies. The main strengths consistently highlighted were the clean experimental control (disentangling interference from context length), the consistency of results across many models, and the clear empirical pattern of “old-value” errors.

However, several concerns moderated enthusiasm and informed a borderline decision:

- Novelty relative to prior work:
Multiple reviewers questioned whether PI-LLM reveals a genuinely new phenomenon or primarily reframes known RAG/inter-context conflict effects. Early versions were seen as insufficiently positioned relative to existing RAG conflict, noise-in-context, and MRCR-style benchmarks.

- Ecological validity and realism:
The synthetic key–value setup was viewed as overly abstract, raising doubts about how directly the findings transfer to realistic long-context use cases (e.g., dialogue, document editing, or agents).

- Actionability and mitigation:
Reviewers expressed concern that the paper mostly demonstrates failure without offering concrete training-time or architectural solutions, beyond showing that prompting and CoT are insufficient.

- Presentation and framing:
Some reviewers found the manuscript initially hard to read, with cluttered figures and an unclear framing (benchmark vs. cognitive science vs. interpretability).

Overall, while reviewers acknowledged the importance and consistency of the phenomenon, the paper sat near the acceptance boundary due to questions about incremental contribution versus explanatory depth, and about how much it advances practical understanding beyond existing long-context evaluations.

**Reviewer Concerns:**

Largely Addressed by the Rebuttal

- Positioning and novelty

The authors substantially clarified that PI-LLM is not a RAG benchmark, but a cognitive-science–grounded working-memory probe focused on multi-binding interference.

- Related work was expanded to explicitly situate PI-LLM relative to RAG conflict studies, GSM-Infinite, and MRCR.

The connection to human proactive interference paradigms and to entity-binding interpretability helped distinguish the contribution as explanatory rather than purely evaluative.

- Mechanistic depth

The newly added PCA-based mechanistic analysis linking behavioral failure to compression of a binding subspace directly addressed requests for deeper insight and moved the paper beyond surface-level benchmarking.

- Ecological validity

Added experiments on naturalistic meeting transcripts and explicit linkage to MRCR results from Gemini and GPT-5 reports addressed concerns that the task was purely toy-like.

These results convincingly showed that the same error pattern (retrieval of outdated bindings) appears outside synthetic lists.

- Mitigation via prompting and agents

The rebuttal thoroughly demonstrated that aggressive prompting, high-effort reasoning, and contemporary agentic frameworks yield only partial, curve-shifting improvements, not elimination of the interference pattern—addressing reviewers who speculated that the issue might be easily solvable with current tools.

Partially or Still Outstanding

- Actionable solutions

While the mechanistic analysis suggests directions (e.g., preserving multi-binding subspaces), reviewers seeking concrete architectural or training-time interventions may still find the paper diagnostic rather than prescriptive.

- Generality beyond binding-heavy tasks

Although realism concerns were mitigated, some reviewers may remain unconvinced that PI-LLM captures most long-context failures, rather than a particularly severe but specific subclass (repeated rebinding of the same entity).

- Presentation polish

Minor concerns about figure clarity and overall presentation quality remain, though these are largely fixable in revision.

**Reviewer Scores:**

Reviewer LZv7 (initially ~4)
Likely stable. The clarified novelty, explicit distinction from RAG, and mechanistic analysis directly address their core skepticism, though concerns about practicality may persist.

Reviewer zn3J (initially ~6)
Likely stable. Their main requests—trend across stronger models, related work, and insight into mechanisms—were directly answered.

Reviewer eEkx (initially ~4)
Likely stable. The added mechanistic explanation, agentic results, and naturalistic validation address most objections, though they may still desire stronger “how to fix it” conclusions.

Reviewer Ue62 (initially ~4)
Likely stable. The ecological-validity experiments and tokenizer clarification address key weaknesses, but the task may still feel narrow.

---

### Decision · Program_Chairs · 2026-01-26

Reject